# A pair of dopaminergic neurons DAN-c1 mediate *Drosophila* larval aversive olfactory learning through D2-like receptors

**Cheng Qi, Cheng Qian†, Emma Steijvers‡, Robert A Colvin, Daewoo Lee***

Department of Biological Sciences, Ohio University, Athens, United States

**\*For correspondence:**
leed1@ohio.edu

**Present address:** †School of Medicine, Johns Hopkins University, Baltimore, United States; ‡Department of Physiology, Development and Neuroscience, University of Cambridge, Cambridge, United Kingdom

**Competing interest:** The authors declare that no competing interests exist.

## eLife Assessment

This study presents **valuable** findings on the role of dopamine receptor D2R in dopaminergic neurons DAN-c1 and mushroom body neurons (Y201-GAL4 pattern) on aversive and appetitive conditioning. The evidence supporting the claims of the authors is **solid** in the context of their behavioural paradigm. Controls using a reciprocal training protocol would have broadened the scope of their conclusions. The work will be of interest to researchers studying the role of dopamine during learning and memory.

**Abstract** The intricate relationship between the dopaminergic system and olfactory associative learning in *Drosophila* has been an intense scientific inquiry. Leveraging the formidable genetic tools, we conducted a screening of 57 dopaminergic drivers, leading to the discovery of DAN-c1 driver, uniquely targeting a pair of dopaminergic neurons (DANs) in the larval brain. While the involvement of excitatory D1-like receptors is well established, the role of D2-like receptors (D2Rs) remains underexplored. Our investigation reveals the expression of D2Rs in both DANs and the mushroom body (MB) of third-instar larval brains. Silencing D2Rs in DAN-c1 via microRNA disrupts aversive learning, further supported by optogenetic activation of DAN-c1 during training, affirming the inhibitory role of D2R autoreceptor. Intriguingly, D2R knockdown in the MB impairs both appetitive and aversive learning. These findings elucidate the distinct contributions of D2Rs in diverse brain structures, providing novel insights into the molecular mechanisms governing associative learning in *Drosophila* larvae.

## Introduction

Learning defines a behavioral change that results from acquiring information about the environment, and memory refers to the process by which the information is encoded, stored, and later retrieved. Learning and memory form the basis for higher brain functions, including cognition and decision-making, which shapes our individuality (*Kandel et al., 2014*). On the cellular and physiological level, learning and memory are achieved by neuroplastic changes in circuits, including neuronal excitability and synaptic plasticity. Usually, distinct types of neurotransmitters, such as dopamine (DA), modulate these changes.

DA plays an important role in many mammalian brain functions, including motor functions, motivation, reinforcement, addiction, and learning and memory (*Neve et al., 2004*; *Baik, 2013*; *Missale et al., 1998*). Dopaminergic neurons (DANs) are mainly located in the mesencephalon: DANs in the substantia nigra are responsible for motor functions, while those in the ventral tegmental area are

important in reward, addiction, and learning and memory (*Baik, 2013*; *Björklund and Dunnett, 2007*). DA achieves its functions via two families of G protein-coupled receptors: excitatory D1-like and inhibitory D2-like receptors (D2Rs; *Missale et al., 1998*). All D1-like receptors are located postsynaptically; in contrast, D2Rs both function postsynaptically and play an important presynaptic role, regulating DA release through negative feedback (*Missale et al., 1998*). All these receptors are important in mammalian associative learning (*Puig et al., 2014*; *Barrot, 2014*). D1-like receptors elevate intracellular cAMP by activating adenylyl cyclase (AC) via $G\alpha_s$, while D2Rs repress cAMP by inhibiting AC via $G\alpha_{i/o}$. cAMP activates protein kinase A (PKA), leading to the phosphorylation of DARPP-32 (dopamine and cyclic AMP-regulated phosphoprotein, 32 kDa), ion channels, and CREB (cAMP response element-binding protein). In addition, DA receptors also activate the PLC-PKC, MAPK, and CaMKII pathways (*Neve et al., 2004*; *Baik, 2013*; *Missale et al., 1998*; *Savica and Benarroch, 2014*; *Beaulieu and Gainetdinov, 2011*; *Klein et al., 2019*).

Although mammalian studies reveal mechanisms more relevant to human beings, the complexity of the nervous system impedes our understanding of the basic or more universal principles of learning and memory applicable generally to all nervous systems (*Waddell, 2010*). With a simple central nervous system (CNS) and powerful genetic tools, the fruit fly *Drosophila melanogaster* has become a popular model organism in learning and memory research (*Berry et al., 2008*; *Bellen et al., 2010*). With conserved genes in DA metabolism and signaling (*Yamamoto and Seto, 2014*), as well as fundamental similarities in the olfactory circuitry compared to mammals (*Heisenberg, 2003*; *Davis, 2004*), *Drosophila* can perform olfactory associative learning in both larvae (*Honjo and Furukubo-Tokunaga, 2005*; *Honjo and Furukubo-Tokunaga, 2009*; *Widmann et al., 2016*) and adults (*Quinn et al., 1974*; *Tempel et al., 1983*; *Roman and Davis, 2001*; *Quinn and Dudai, 1976*; *Waddell et al., 2000*). Olfactory associative learning is a type of classical conditioning in which flies are trained under positive or negative reinforcement paired with an odorant. Different from the naïve reaction to the odorant, flies approach the odorant after being trained with rewards (e.g. sucrose [SUC]; appetitive) (*Tempel et al., 1983*), but avoid the odorant when trained with punishments (e.g. electric shock, bitter taste chemicals; aversive) (*Quinn et al., 1974*). Several genes related to the cAMP-PKA signaling pathway, including *dunce* (*dnc*) (*Dudai et al., 1976*) and *rutabaga* (*rut*) (*Tempel et al., 1983*), are expressed in the mushroom body (MB, a center for *Drosophila* learning and memory) in both larval and adult brains (*Davis and Dauwalder, 1991*; *Crittenden et al., 1998*). Mutations of these genes lead to learning deficiencies, indicating their roles in larval (*Honjo and Furukubo-Tokunaga, 2005*; *Honjo and Furukubo-Tokunaga, 2009*) and adult olfactory learning (*Tempel et al., 1983*; *Dudai et al., 1976*).

*Drosophila* larvae offer several advantages for studying olfactory learning compared to adults. Notably, compared to neural circuits underlying olfactory learning, their simpler neural circuitry (*Saumweber et al., 2018*), characterized by fewer olfactory receptor neurons (*Stocker, 2001*), projection neurons (PNs) (*Ramaekers et al., 2005*), mushroom body neurons (MBNs) (*Lee et al., 1999*), and DANs (*Blanco et al., 2011*), facilitates the elucidation of underlying mechanisms. Additionally, larvae exhibit simpler behavioral patterns, facilitating experimental manipulations and observations. Furthermore, their translucent cuticles enable convenient application of techniques such as optogenetics (*Schroll et al., 2006*), further enhancing the experimental versatility of larval studies.

Like in mammalian brains, DA achieves its functions via four DA receptors in flies, two D1-like receptors dDA1 (*Sugamori et al., 1995*) (or Dop1R1) and DAMB (*Han et al., 1996*) (or Dop1R2), one D2R (*Hearn et al., 2002*) (or Dop2R), and one non-canonical receptor DopEcR (*Yamamoto and Seto, 2014*; *Srivastava et al., 2005*). dDA1 is mainly found in the MB (*Kim et al., 2003*; *Selcho et al., 2009*) and is necessary for appetitive and aversive olfactory learning in larvae (*Selcho et al., 2009*) and adults (*Kim et al., 2007*). D2R in the adult MB is necessary for anesthesia-resistant memory (*Scholz-Kornehl and Schwärzel, 2016*). In addition, D2R in GABAergic anterior paired lateral neurons is known to secure aversive conditioning in adult flies (*Zhou et al., 2019*). Although D2R expression has been reported in the ventral nerve cord (*Draper et al., 2007*), neither its expression in larval brains nor its functions in larval olfactory learning have been investigated.

By using a GFP-tagged D2R strain, we detected the expression pattern of D2R in the third-instar larval brain, specifically in DANs and MBNs. Knockdown of D2Rs in a pair of DANs, DAN-c1, impaired aversive learning, while knockdown of D2R in MBNs led to deficits in both aversive and appetitive learning. These results revealed that D2Rs in distinct brain structures mediate different learning tasks.

The newly discovered roles of D2R in the larval brain provide new insights into the mechanisms underlying larval associative learning.

## Results

### Distinct DANs innervate different compartments of the MB

The connectome of larval learning circuitries has been investigated in both first- and third-instar larvae (*Saumweber et al., 2018*; *Eichler et al., 2017*). The MB serves as a primary learning center in *Drosophila*, which is composed of αβ, α′β′, and γ neurons in adult brains (*Tanaka et al., 2008*). In larvae, axons from γ neurons bifurcate and form the vertical and medial lobes, as αβ and α′β′ neurons are not mature (*Lee et al., 1999*; *Kunz et al., 2012*; *Kurusu et al., 2002*). These lobes are divided into 11 compartments (refer to *Figure 1*) based on the coverage of neurites from both MB extrinsic neurons and MB output neurons (MBONs) (*Saumweber et al., 2018*). Around 21 DANs are found in each third-instar brain hemisphere and categorized into four clusters: DM1 (dorsomedial), pPAM (primary protocerebral anterior medial, or DM2) (*Rohwedder et al., 2016*), DL1 (dorsolateral), and DL2 (*Blanco et al., 2011*). DL1 neurons project to the vertical lobe (*Selcho et al., 2009*), while primary protocerebral anterior medial (pPAM) neurons innervate the medial lobe (*Rohwedder et al., 2016*) (refer to *Figure 1a–c*).

In this study, we wanted to functionally identify individual DANs that mediate larval olfactory learning. The first step was to search for DAN-specific driver strains that mark a few DANs, which subsequently can be used to target genetic manipulations of corresponding neurons. A total of 57 driver strains identifying DANs were screened in this study (*Table 1*). These strains were chosen based on previous studies, either identifying a pair of DANs in larvae or identifying only several in adult brains and indicating the potential of identifying a pair of DANs in larvae. TH-GAL4, a traditional dopaminergic neuronal driver (*Friggi-Grelin et al., 2003*), identifies all DANs except those in pPAM (*Figure 1d and a*). Split-GFP reconstitution across synaptic partners (GRASP) technique was used to investigate the 'direct' synaptic connections from DANs to the MB, in which portions of GFP were specifically expressed in corresponding neurons (*Figure 1—figure supplement 2d*; *Macpherson et al., 2015*). GRASP results showed neurons under TH-GAL4 formed synapses in the vertical lobe and lower peduncle (LP) (white dashed lines in *Figure 1d′*), consistent with previous electron microscopy data (*Eichler et al., 2017*). We found three DAN driver strains that identify a pair of DANs in the third-instar larval brain. DAN driver R76F02-AD;R55C10-DBD identifies a pair of DANs innervating the LP, which would be DAN-c1 based on previous published nomenclature (*Saumweber et al., 2018*; *Figure 1f*). MB296B driver identifies the dopaminergic neurons (DAN-d1) projecting to the lateral appendix (LA) (*Figure 1g*), as well as many non-DANs. SS1716 driver identifies a pair of DANs forming synapses in the lower vertical lobe (LVL), indicating it is DAN-g1 (*Figure 1h*).

In pPAM, R58E02 driver identifies DAN-h1, i1, and j1, innervating the shaft (SHA), intermediate toe (IT), and upper toe (UT) (*Figure 1i*), and R30G08 driver identifies DAN-h1 and i1 (*Figure 1j*). As described in a previous report (*Saumweber et al., 2018*), SS864 driver identifies DAN-i1, innervating the upper toe (*Figure 1l*), and SS1757 driver identifies DAN-k1 which innervate the lower toe (LT) (*Figure 1m*). In contrast, the SS1696 driver identifies not only DAN-h1, but also i1 and one DL1 neuron not innervating the MB (*Figure 1k*).

In summary, our results show that five DL1 and four pPAM DANs innervate nine distinct MB compartments in a one-to-one pattern (*Figure 1b and c*). DL1 neurons innervate the vertical lobe and peduncle, while pPAM neurons project to the medial lobe. The neuronal driver strains screened can be used to investigate the roles of individually identified DANs in larval olfactory learning.

### Dopamine release from DAN-c1 mediates larval aversive learning

Dopamine plays an important role during olfactory associative learning in both adults (*Waddell, 2013*; *Busto et al., 2010*) and larvae (*Honjo and Furukubo-Tokunaga, 2005*; *Honjo and Furukubo-Tokunaga, 2009*). In adults, DANs in PPL1 regulate aversive learning (*Aso and Rubin, 2016*; *Schwaerzel et al., 2003*; *Aso et al., 2012*; *Masek et al., 2015*), while those in protocerebral anterior medial (PAM) mediate reward signals in appetitive learning (*Liu et al., 2012*; *Burke et al., 2012*; *Yamagata et al., 2015*). In larvae, DL1 neurons innervating the vertical lobe and the peduncle are required for aversive

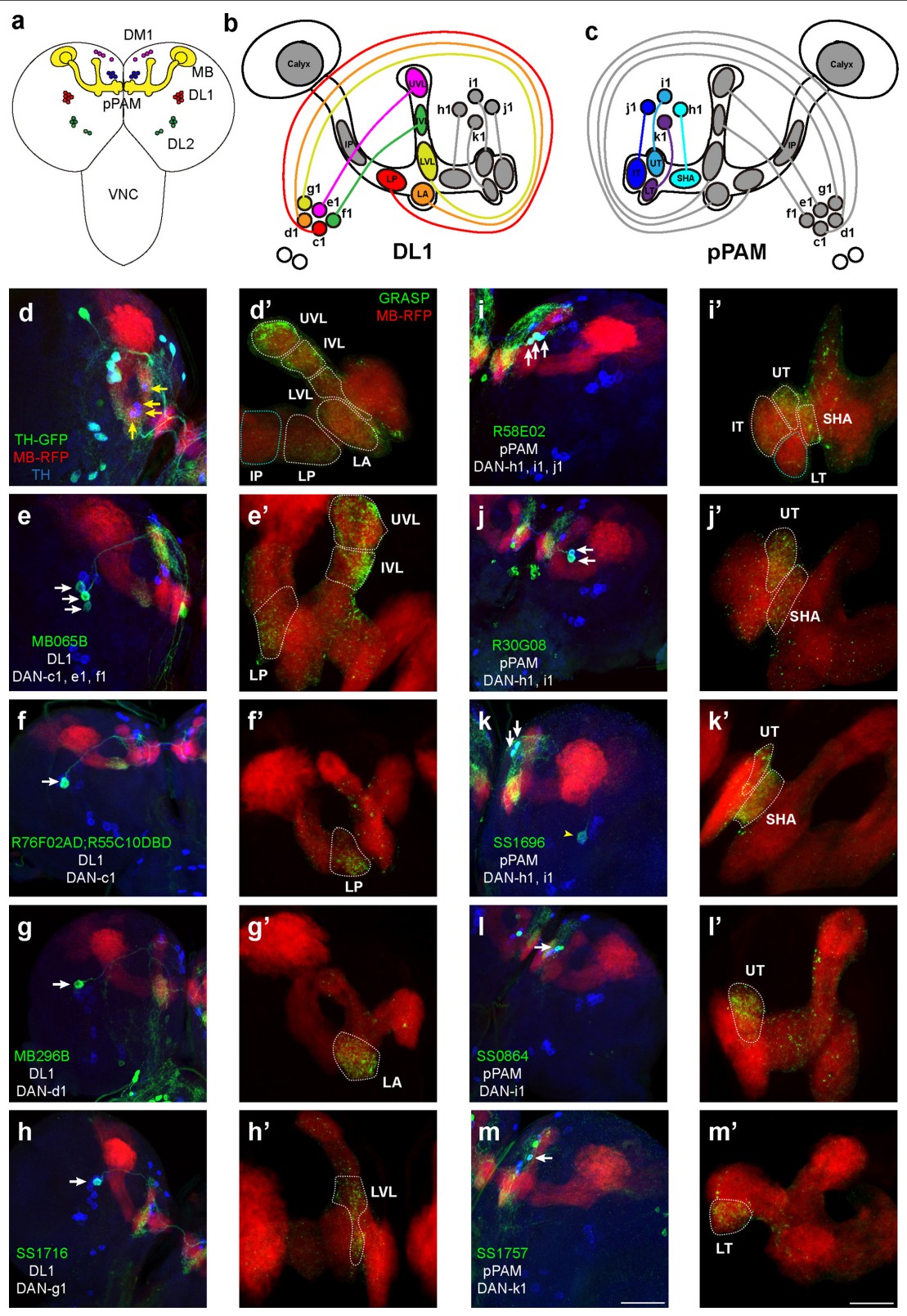

**Figure 1.** Identification of driver strains for a pair of dopaminergic neurons (DANs) in the *Drosophila* larval brain. (**a**) A schematic diagram shows the DAN clusters and the mushroom body (MB) in the third-instar larval brain. (**b** and **c**) Schematic diagrams show the innervation patterns from distinct DANs to different compartments in the MB. All 11 MB compartments are shown; note that there is no synapse formed from DANs to the calyx and intermediate peduncle (IP). DANs in DL1 (**b**) and in pPAM (**c**). (**d–m**) Drivers covering DANs in the DL1 cluster (**d–h**). Drivers covering DANs in the pPAM

*Figure 1 continued on next page*

*Figure 1 continued*

cluster (**i–m**). The first column shows drivers covering distinct DANs. Neurons under the drivers are labeled by GFP, the MB is labeled by RFP, and DANs are marked by tyrosine hydroxylase (TH) antibody. The second column (**d'–m'**) shows the GFP reconstitution across synaptic partners (GRASP) signals from DANs under the driver to the corresponding compartments in the MB. The green channel represents the GRASP signals, and RFP marks the morphology of the MB. In the first column, white arrows mark the DANs under the driver strains. Yellow arrows in (**d**) show the pPAM neurons not labeled by TH-GAL4 driver strain. The yellow arrowhead in (**k**) showed the DL1 neuron not innervating the MB. Scale bars: 50 μm for the first column, and 20 μm for the second column. *Abbreviations:* DL, dorsolateral; DM, dorsomedial; IP, intermediate peduncle; IT, intermediate toe; IVL, intermediate vertical lobe; LA, lateral appendix; LP, lower peduncle; LT, lower toe; LVL, lower vertical lobe; pPAM, primary protocerebral anterior medial; SHA, shaft; UT, upper toe; UVL, upper vertical lobe. *Note* GFP expression patterns in the entire larval central nervous system (CNS) by GAL4 driver strains used in this study can be found in *Figure 1—figure supplement 1*. N numbers for each strain can be found in *Supplementary file 2*.

The online version of this article includes the following figure supplement(s) for figure 1:

**Figure supplement 1.** GFP expression patterns in the larval central nervous system (CNS) by GAL4 driver strains used in this study.

**Figure supplement 2.** Controls for GFP reconstitution across synaptic partners (GRASP) experiments.

learning (*Honjo and Furukubo-Tokunaga, 2009*; *Selcho et al., 2009*), while those in pPAM projecting to the medial lobe are involved in appetitive learning (*Rohwedder et al., 2016*).

In *Figure 1* and *Table 1*, three driver strains identifying distinct pairs of DANs in DL1 were discovered, which could be candidates to investigate their roles in larval aversive learning. The R76F02-AD;R55C10-DBD strain identifies MB-MP1 in the adult brain (*Xie et al., 2018*), which is a DAN involved in adult aversive learning (*Aso et al., 2010*). Based on the analysis with 22 brain samples, we observed this driver strain labels one neuron per hemisphere in the third-instar larval brain (*Figures 2a–d and 1c*, *Supplementary file 3*). Using a UAS-DenMark;UAS-sytGFP strain, its dendrites were labeled with RFP and axonal terminals were marked by GFP. Its dendrites were localized in the dorsomedial protocerebrum (dml), and its axonal terminals located in the LP of the MB (*Figure 2a and c*), with GRASP results supporting the existence of synapses in this compartment (*Figure 1f'*). All these characteristics are consistent with the previously published nomenclature (*Saumweber et al., 2018*), indicating that this pair of neurons is DAN-c1 (*Figure 2d*); thus, this strain will now be referred to simply as DAN-c1.

To reveal the role of DAN-c1 in larval olfactory learning, a single odor learning paradigm and thermogenetic tools were applied (*Honjo and Furukubo-Tokunaga, 2005*; *Honjo and Furukubo-Tokunaga, 2009*; *Qi and Lee, 2014*). Compared to those trained with distilled water (DW), control strains of larvae exhibited repulsion to the odorant pentyl acetate (PA) after being trained with quinine (QUI) paired with PA, reflecting aversive learning. In contrast, larvae were attracted to PA after being trained with sucrose (SUC) paired with PA, reflecting appetitive learning. The extent of repulsion or attraction was represented with a response index (R.I.) that is compared to the DW group. (*Figure 2e*; For further details, refer to the Materials and methods section).

To examine the role of DAN-c1 in aversive learning, we used a *Shibire*[ts1] strain, which encodes a thermosensitive mutant of dynamin blocking neurotransmitter release when the ambient temperature is higher than 30°C by repressing endocytosis and vesicle recycling functions (*Honjo and Furukubo-Tokunaga, 2005*). When trained with QUI at 34°C, larvae with *Shibire*[ts1] expression in DAN-c1 showed significantly increased R.I. compared to that at room temperature (22°C), while it is not significantly different from the DW at 34°C group (*Figure 2f*). The complete inactivation of dopamine release from DAN-c1 with *Shibire*[ts1] impaired aversive learning, indicating that dopamine release from DAN-c1 is important for larval aversive learning to occur.

In the next experiments, a fly strain carrying temperature-sensitive cation channel, dTRPA1, was used to excite the DAN-c1 neuron because it can be activated at temperatures higher than 30°C (*Masek et al., 2015*). Compared to those at 22°C, activation of DAN-c1 with dTRPA1 at 34°C during training induced repulsion to PA in the DW group, while it is not significantly different from the QUI group at 22°C (*Figure 2i*). These data suggested that DAN-c1 excitation, and presumably increased dopamine release, leads to larval aversive learning in the absence of gustatory pairing.

Combining the blockade results with *Shibire*[ts1], these data revealed that dopamine released from DAN-c1 activation mediates larval aversive learning. However, when paired with a gustatory stimulus (QUI or SUC), activation of DAN-c1 during training impairs both aversive and appetitive learning (*Figure 2i*). We suggest that these data indicate a critical role for the amount of dopamine release from DAN-c1 in larval associative learning, as dTRPA1 stimulation may result in excessive dopamine release (see the Discussion section).

**Table 1.** Driver strains screened for dopaminergic neurons in the third-instar larval brain.

Strains are listed with their published names and names used in this work. The numbers of dopaminergic and non-dopaminergic neurons in the third-instar larval brain are described. The identities of dopaminergic neurons from distinct clusters are also listed, especially for those in DL1 and primary protocerebral anterior medial (pPAM). The analogs column lists the labeled neurons in previous publications. Source/gift column shows the original papers in which these strains were described, as well as the laboratories these strains were obtained from. Several, 2–5 neurons; some, 6–10 neurons; lots, >10 neurons.

| | Strains | Name in this work | Neurons in third-instar larval brains | | | | |
| | | | DANs | Non-DANs | Analogs | Stock # | Source/Gift |
|---|---|---|---|---|---|---|---|
| 1 | TH-Gal4 | – | All DANs except pPAM | Some weak | All DANs except PAM (adult) | – | Dr. J. Hirsh's Lab |
| 2 | TH-C' | – | 1 DL1, 1 DM1, 3DL2a | Rare | PPL1 +PPM3 (adult) | – | Dr. M. Wu's lab (*Liu et al., 2012*) |
| 3 | TH-C1 | – | – | – | PPL1 +PPM3 (adult) | – | |
| 4 | TH-D' | – | 3DL1, 2DL2b | Rare | PPL1 +PPM3 (adult) | – | |
| 5 | TH-D1 | – | – | – | PPL1 +PPM3 (adult) | – | |
| 6 | TH-D4 | – | – | – | PPL1 +PPM3 (adult) | – | |
| 7 | TH-F1 | – | 3DL1 | Rare | PPL1 +PPM3 (adult) | – | |
| 8 | Th-F2 | – | 3 DM1, 2–4 DL1, 2DL2a | – | PPL1 +PPM3 (adult) | – | |
| 9 | Th-F3 | – | 3 DM1b, 1 DL1 | – | PPL1 +PPM3 (adult) | – | |
| 10 | TH-G1 | – | 2–3 DM1, 2–4 DL1, 2 DL2b | – | PPL1 +PPM3 (adult) | – | |
| 11 | R30G08 | – | h1, i1, weak k1 | – | 2 pPAM (L3) | 48101 | BDSC (*Rohwedder et al., 2016*) |
| 12 | R58E02 | – | h1, i1, j1 | 1 | 3 pPAM (L3) | 41347 | |
| 13 | R64H06 | – | h1, i1, j1, k1 | Lots | 4 pPAM (L3) | 49608 | |
| 14 | MB315C | – | 1–3 pPAM (h1) | Lots | PAM-γ5 (adult) | – | Janelia Farm (*Aso et al., 2014a*) |
| 15 | MB109B | – | – | – | PAM-β'2a, PAM-γ5 (adult) | – | |
| 16 | MB301B | – | – | – | PAM-β'2m, PAM-β2β'2a (adult) | – | |
| 17 | MB056B | – | – | – | PAM-β'2m, PAM-β'2p (adult) | – | |
| 18 | MB032B | – | – | – | PAM-β'2m, PAM-β'2p, PAM-β2β'2 a, PAM-γ3 (adult) | – | |
| 19 | MB312B | – | – | – | PAM-γ4, PAM-γ4<γ1γ2 (adult) | – | |
| 20 | MB194B | – | – | – | PAM-α1, PAM-β'2a, PAM-β1, PAM-β1ped, PAM-β2 (adult) | – | |
| 21 | MB063B | – | – | – | PAM-β1 (adult) | – | |
| 22 | MB043B | – | h1, i1, j1 | Weak, lots of MBN | PAM-α1, PAM-β'1 ap, PAM-β'1m, PAM-β1 (adult) | – | |
| 23 | MB441B | – | – | – | PAM-γ3 (adult) | – | |
| 24 | MB025B | – | – | – | PAM-β'1ap/m (adult) | – | |
| 25 | MB438B | – | – | – | PPL1-α'2α2, PPL1-α3, PPL1-γ1pedc (adult) | – | |
| 26 | MB296B | DAN-d1 | d1 | Some in VNC | PPL1-γ2α'1 (adult) | – | |
| 27 | MB304B | – | – | – | PPL1-α'3 (adult) | – | |
| 28 | MB058B | – | – | – | PPL1-α'2α'2 (adult) | – | |
| 29 | MB065B | – | c1, f1, e1 | – | PPL1-α'2α2, PPL1-α'3, PPL1-α3, PPL1-γ2α'1 (adult) | – | |

*Table 1 continued on next page*

*Table 1 continued*

| | Strains | Name in this work | Neurons in third-instar larval brains | | | Stock # | Source/Gift |
|---|---|---|---|---|---|---|---|
| | | | DANs | Non-DANs | Analogs | | |
| 30 | GMR_SS01716 | DAN-g1 | g1 | – | g1 (LVL) (L3) | – | Dr. M. Zlatic's Lab (*Saumweber et al., 2018*) |
| 31 | GMR_SS01696 | DAN-h1 | h1 +i1 | – | h1 (SHA) (L3) | – | |
| 32 | GMR_SS00864 | DAN-i1 | i1 | 1 | i1 (UT) (L3) | – | |
| 33 | GMR_SS1757 | DAN-k1 | k1 | 1 | k1 (LT) (L3) | – | |
| 34 | R78E04 | – | 1 weak DL1 | Several | d1 (LA) (L3) | 39997 | |
| 35 | R72B05 | – | 1 DL1 | Lots | MBIN-e2 (L3) | 39611 | |
| 36 | R12C11 | – | 2 DM1b | Several | MBON-c2 (L3) | 76324 | |
| 37 | R30F04 | – | weak d1 | Rare | d1 (LA) (L3) | 48614 | |
| 38 | R76C04 | – | c1, d1 | Rare | c1 (LP) (L3) | 48621 | |
| 39 | R37D06 | – | – | – | f1 (IVL) (L3) | 47921 | |
| 40 | R14E06 | – | 2 DL1 | Several | MBIN-e2 (L3) | 48643 | |
| 41 | TH-C-AD;R76F05-DBD | – | 1 weak DL1, 2 strong DM1 | Lots | PPL1 (SMP-PED) (adult) | – | Dr. M. Wu's Lab (*Xie et al., 2018*) |
| 42 | TH-F-AD;R61H03-DBD | – | 2DL2, 3 DM1 | Rare | PPL1 (SMP) (adult) | – | |
| 43 | R76F02-AD;TH-F-DBD | – | c1, e1 | Lots weak | PPL1 (MB-MP1) (adult) | – | |
| 44 | TH-F-AD;R76F05-DBD | – | 1 weak DL1, 1–2 strong DM1 | Several | PPL1 (SMP-γ) (adult) | – | |
| 45 | R76F02-AD;R60F07-DBD | – | c1, f1, e1 | Rare | PPL1 (PED, MB-MP1) (adult) | – | |
| 46 | R60F07-AD;R76F05-DBD | – | g1, e1 | Strong SOG | PPL1 (SMP-γ) (adult) | – | |
| 47 | DAT-B-AD;R76F05-DBD | – | 2-3pPAM, 1 DL1 | Rare | PPL1 (hSMP-γ) (adult) | – | |
| 48 | R76F02-AD;R55C10-DBD | DAN-c1 | c1 | – | PPL1 (MB-MP1) (adult) | – | |
| 49 | R76F02-AD;R76F01-DBD | – | 1 weak DL1 | Lots | PPL1 (dFB) (adult) | – | |
| 50 | R76F02-AD;R76F05-DBD | – | – | Lots | PPL1 (dFB) (adult) | – | |
| 51 | R76F05-AD;R61H03-DBD | – | 1 DL1, 2 pPAM, 3 DM1 | 1 | PPL1 (MB-MV1) (adult) | – | |
| 52 | WED-1 | – | – | – | WED (adult) | – | Dr. M. Wu's Lab (*Liu et al., 2012*) |
| 53 | WED-2 | – | – | – | WED (adult) | – | |
| 54 | TH-FLP/UAS-FRT>>FRT-CD8::GFP;R39G05 | – | – | – | PPL1 (bSMP-γ) (adult) | 50063 | |
| 55 | TH-FLP/UAS-FRT>>FRT-CD8::GFP;R17C11 | – | – | – | PPL1 (V1 and MP1) (adult) | 48763 | |
| 56 | TH-FLP/UAS-FRT>>FRT-CD8::GFP;R39C07 | – | 2 DL1 | – | PPL1 (MV1 and MP1) (adult) | 50039 | |
| 57 | TH-FLP/UAS-FRT>>FRT-CD8::GFP;R67E08 | – | – | – | PPL1 (V1) (adult) | 39445 | |

## The expression pattern of D2R in the third-instar larval brain

Although dopamine D1-like receptors have been proven important for learning (*Kim et al., 2007*), the role of D2Rs has not been fully investigated. In addition, the expression pattern of D2R in fly brains was not reported. A fly strain expressing GFP-tagged D2R (BDSC#60276) was used to reveal the expression pattern of D2R in the third-instar larval brain (*Figure 3a*). D2Rs were found in DANs and the MB. In DANs (*Figure 3b–g*), D2Rs were found in DM1, pPAM, DL2b, and some DL1 neurons. In the MB, D2Rs were expressed in the soma and lobes, but not in the calyx (*Figure 3h and i*). Even though D2Rs were widely found in vertical lobes, medial lobes, and peduncles, they were not expressed

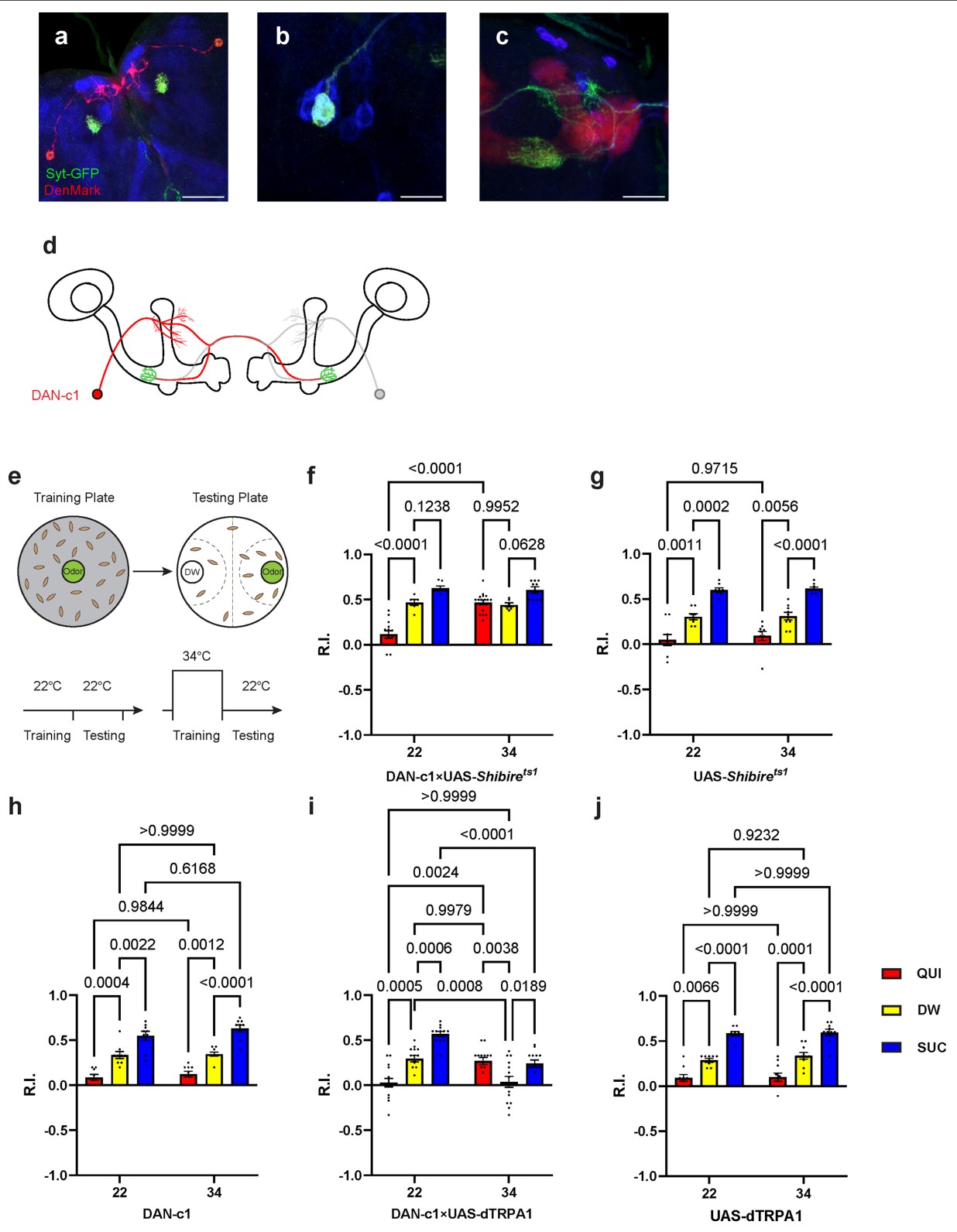

**Figure 2.** Dopamine release from DAN-c1 mediates larval aversive learning. (**a–d**) R76F02-AD;R55C10-DBD driver is used as it covers one dopaminergic neuron, DAN-c1, in each brain hemisphere. (**a**) Dendrites and axons of DAN-c1 are labeled by DenMark and sytGFP, correspondingly. (**b and c**) Soma and neurites from *Figure 1f* with higher magnification. DAN-c1 is labeled with GFP, the mushroom body (MB) with RFP, and dopaminergic neurons (DANs) with TH antibody (blue color). Only one DAN soma is identified (**b**), axons from DAN-c1 innervate the lower peduncle (LP) of the MB (**c**). A

*Figure 2 continued on next page*

*Figure 2 continued*

schematic diagram (**d**) shows the innervation patterns of DAN-c1. Modified from *Eichler et al., 2017* and *Saumweber et al., 2018*. (**e**) A schematic paradigm for larval olfactory learning (top) and two different training paradigms for thermogenetics (bottom). (**f–h**) Blocking dopamine release from DAN-c1 during learning using *shibire*[ts1] strain at 34°C impairs larval aversive learning. (**h–j**) Activation of DAN-c1 with dTRPA1 at 34°C induces aversive learning. QUI, quinine; DW, distilled water; SUC, sucrose. Data are shown as mean ± SEM. Two-way ANOVA, Tukey's multiple comparison test. For N numbers, interaction p-values, row factor p-values, and column factor p-values, see *Supplementary file 4*. Scale bars: 50 μm (**a**), 20 μm (**b and c**). *Note*: N numbers of immunostaining for each strain can be found in *Supplementary file 2*.

in every MBN. A transection of the peduncle region showed the absence of D2Rs in the core area (*Figure 3—figure supplement 1i*), which is composed of densely packed newly created fibers and lacks Fasciclin II (FAS II) (*Kurusu et al., 2002*).

To inspect whether the pattern of GFP signals indeed reflected the expression of D2R, three D2R enhancer driver strains (R72C04, R72C08, and R72D03-GAL4) were crossed with the GFP-tagged D2R strain. R72C08-GAL4 covered three DM1 DANs (*Figure 3—figure supplement 1c*), and R72C04-GAL4 labeled one DM1 and two DL2b DANs (*Figure 3—figure supplement 1d and e*). R72D03-GAL4 identified parts of MBNs, whose axons spread on the surface of the MB lobes (*Figure 3—figure supplement 1f*); R72C08-GAL4 also identified a subset of neurons from four MB neuroblasts, with soma in four clusters and a converged bundle of axons (*Figure 3—figure supplement 1g and h*). These results revealed the expression of D2R in the MBNs and DANs in the third-instar larval brain.

## D2R in DAN-c1 influences larval aversive learning

Our previous work reported that D2R knockdown (UAS-RNAi) in DANs driven by TH-GAL4 impaired larval aversive learning (*Qi and Lee, 2014*). Using a microRNA strain (UAS-D2R-miR) (*Xie et al., 2018*), a similar deficit was observed (*Figure 4—figure supplement 1f*). To further understand the roles of D2R in aversive learning, its expression in distinct DANs, as well as corresponding learning assays, were investigated. Crossing the GFP-tagged D2R strain with a DAN-c1-mCherry strain demonstrated the expression of D2R in DAN-c1 (*Figure 4a*).

To reduce the expression of D2R in DANs, a microRNA strain UAS-D2R-miR was used when crossing with distinct driver strains. The efficiency of D2R knockdown was confirmed by crossing the GFP-tagged D2R strain with TH-GAL4;UAS-D2R-miR strain. In these larval brains, GFP signals in DM1 were not detected, while those in pPAM were still intact (*Figure 4b and c*). Quantification showed a significant decrease of GFP signals in the knockdown group compared to the control (*Figure 4d*), indicating reduced transcripts of D2R linked GFP by D2R-microRNA (*Figure 4—figure supplement 1c*).

To investigate the roles of D2R in distinct DANs during larval associative learning, UAS-D2R-miR strain was crossed with distinct driver strains labeling different pairs of DANs. Among them, the knockdown of D2R in DAN-c1 impaired aversive learning with the odorant PA, while appetitive learning was unaffected (*Figure 4e*). In contrast, although D2R was also found in DAN-d1 and DAN-g1, neither D2R knockdown in DAN-d1 nor in DAN-g1 affected larval olfactory learning (*Figure 4—figure supplement 1d to f*, see the Discussion section). As the naïve sensory and motor functions were not affected, this deficiency was caused by impairment in learning abilities (*Figure 4f–i*). Similar learning deficits were observed in the same strain trained with another odorant, propionic acid (*Figure 4—figure supplement 2a*), as well as in larvae with D2R knockdown using UAS-RNAi (*Figure 4—figure supplement 2b*). These results demonstrated that D2Rs are expressed in DAN-c1, and they are necessary for larval aversive learning. Presumably, the knockdown of presynaptic inhibitory D2R autoreceptors on DAN-c1 will result in increased and excessive dopamine release, which leads to aversive learning deficiency. These results are consistent with the activation studies with dTRPA1 above, in which increased dopamine release during training results in impaired aversive learning (*Figure 2i*).

## Over-excitation of DAN-c1 during learning impairs larval aversive learning

To exclude possible chronic effects of D2R knockdown during development, optogenetics was applied at distinct stages of the learning protocol. Channelrhodopsin2 (ChR2) is a blue light-activated cation channel from algae, which can be used to activate target neurons (*Honjo et al., 2012*; *Boyden et al., 2005*). Over-excitation of DANs under a TH-GAL4 driver with ChR2 during training impaired aversive learning but left appetitive learning intact (*Figure 5—figure supplement 1*), which is consistent

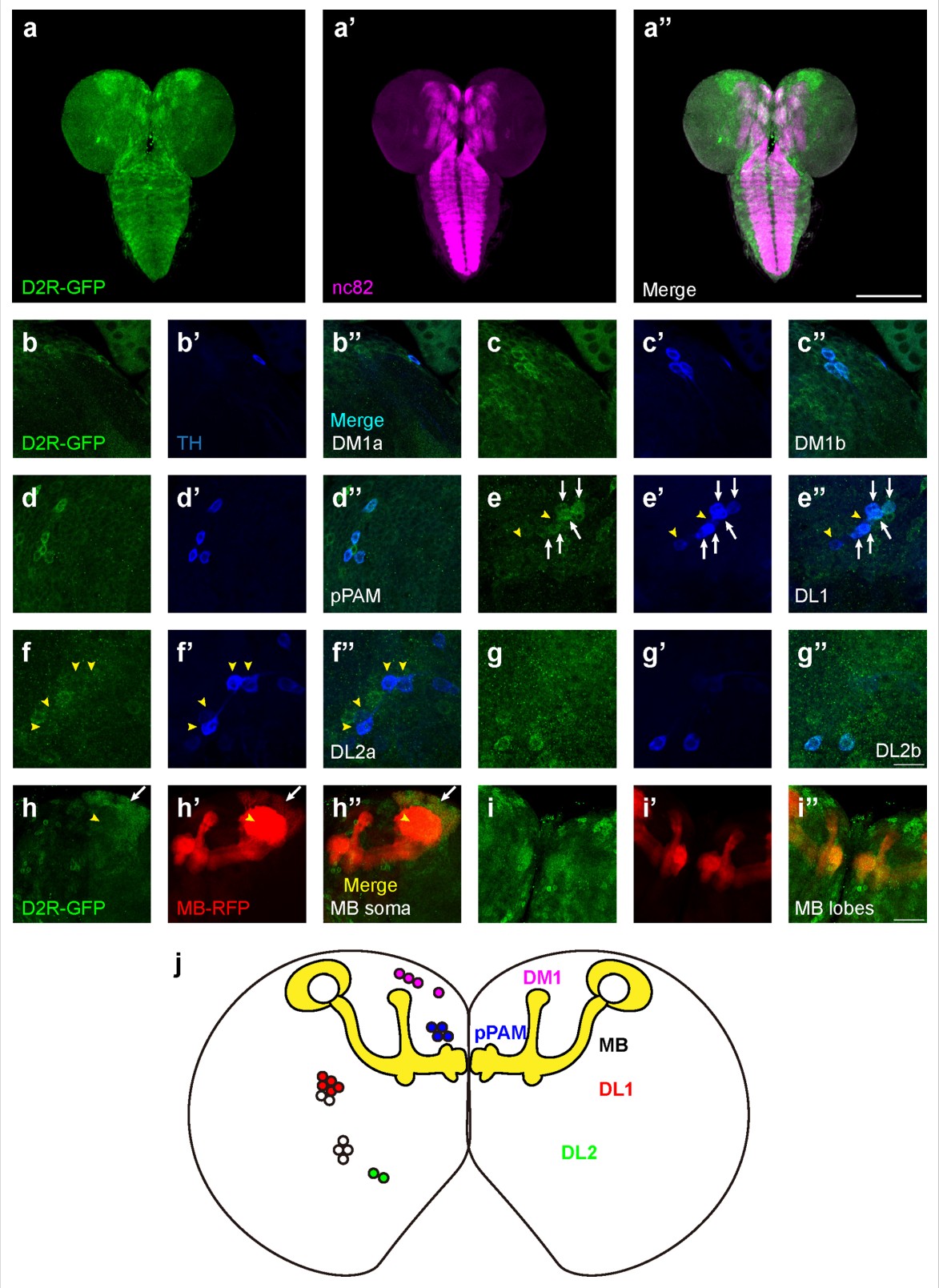

**Figure 3.** D2-like receptors (D2Rs) are expressed in dopaminergic neurons (DANs) and mushroom body (MB) in *Drosophila* larval brains. (**a**) The expression pattern of D2R in a general view. D2R is shown with tagged GFP (D2R-GFP). Magenta represents neuropils marked by nc82 antibody (**a'**). (**b–g**) D2Rs (presynaptic) are found in most DANs: DM1a (**b**) and DM1b (**c**), pPAM (**d**), DL1 (**e**), DL2a (**f**), and DL2b (**g**) clusters. D2Rs are expressed in parts of DL1 neurons (white arrows in **e**) but not in DL2a neurons (yellow arrowheads in **f**). (**h–i**) D2Rs (postsynaptic) are found in the soma of MB neurons (white

*Figure 3 continued on next page*

*Figure 3 continued*

arrows in **h**), and MB lobes and peduncles (**i**), but not in calyx (yellow arrowheads in **h**). (**j**) A schematic diagram shows the expression pattern of pre- and postsynaptic D2R in DANs and MB (yellow) in the *Drosophila* larval brain, respectively. Scale bars: 200 μm (**a**); 50 μm (**b–g**); 20 μm (**h, i**). *Abbreviations:* DL, dorsolateral; DM, dorsomedial; pPAM, primary protocerebral anterior medial. *Note:* N numbers can be found in **Supplementary file 2**.

The online version of this article includes the following figure supplement(s) for figure 3:

**Figure supplement 1.** D2R-GAL4 strains support the expression pattern of D2-like receptor (D2R).

with D2R knockdown results. To investigate the mechanisms with a better temporospatial resolution, ChR2 was expressed in DAN-c1, and blue light was applied at distinct stages of the learning protocol (*Figure 5a*). Optogenetic activation of DAN-c1 during training impaired aversive learning, not appetitive learning (*Figure 5b–d*). This result is consistent with the effect of D2R knockdown in DAN-c1, indicating that increased, excessive dopamine release during training leads to impaired aversive learning.

## D2R in MB mediates larval learning through inhibition

We have shown that D2R in DAN-c1 plays a critical role in larval aversive learning. Since D2Rs are also expressed in soma and axons in most MBNs (*Figure 3h and i*), we examined the role of D2R in MB, a center for learning in *Drosophila*. Knockdown of these D2Rs by D2R-miR impaired both appetitive and aversive learning (*Figure 6a*). Similarly, optogenetic activation of MBNs during training led to deficiencies in both appetitive and aversive learning (*Figure 6b and d*). These deficiencies were not observed in larvae with activation during the resting stage. As D2Rs are inhibitory receptors, and optogenetic activation leads to greater neuronal excitation like what may occur with knockdown of D2Rs, these data show that the inhibitory effect of D2Rs in MBNs is necessary for larval olfactory associative learning to occur.

## Discussion

The dopaminergic system plays an important role in *Drosophila* olfactory associative learning, but the roles of D2R in this process have not been fully explored. In this study, we systematically investigated the expression pattern of D2R in the third-instar larval brain, as well as its role in larval aversive and appetitive learning. One driver strain identifying a pair of DAN-c1 neurons in the third-instar larval brain was discovered, and learning assays with thermogenetic tools (*Shibire^{ts1}*, dTRPA1) demonstrated that the blockade of dopamine release from DAN-c1 impeded aversive learning, while its activation during training led to repulsion toward the odor in the absence of unconditioned stimulus (US) (i.e. QUI). These results revealed that DAN-c1 activation (i.e. presumably leading to the release of synaptic dopamine) mediates larval aversive learning to occur. Subsequently, the expression pattern of D2R was explored by using a GFP-tagged D2R strain, including distinct DANs and MBNs. D2Rs were expressed in DAN-c1, and the knockdown of these receptors resulted in aversive learning deficiency. These data suggested that presynaptic D2Rs in a pair of dopaminergic neurons, DAN-c1, regulate dopamine release during excitation, whereas knockdown of these same receptors leads to excessive dopamine release, causing deficits in aversive learning to occur. Furthermore, the activation of DAN-c1 with optogenetic tools during training, resulting in excessive dopamine release, impaired aversive learning as well. Finally, it was demonstrated that either the knockdown of postsynaptic D2R or activation of MBNs led to learning deficits. These data demonstrate that D2Rs in distinct brain locations are critically involved in associative learning.

### The characteristics of the single odor larval learning paradigm

We adopted the single odor larval learning paradigm from previous publications (*Honjo and Furukubo-Tokunaga, 2005*; *Honjo and Furukubo-Tokunaga, 2009*). To validate this paradigm which induces associative learning responses, Honjo et al. (*Honjo and Furukubo-Tokunaga, 2005*; *Honjo and Furukubo-Tokunaga, 2009*) tested the paradigm from four aspects: First, the paradigm did not show obvious sensitization or habituation effects when larvae were tested. They applied the odorant to the larvae after training. Only the ones who had paired training with both odor and unconditioned stimulus (US, QUI or SUC) showed learning responses. Larvae exposed for 30 min to either the odorant or the US alone did not show a different response to the odor compared to the naïve group. Second, the odor responses are associative. Honjo et al. showed that only when the odorant

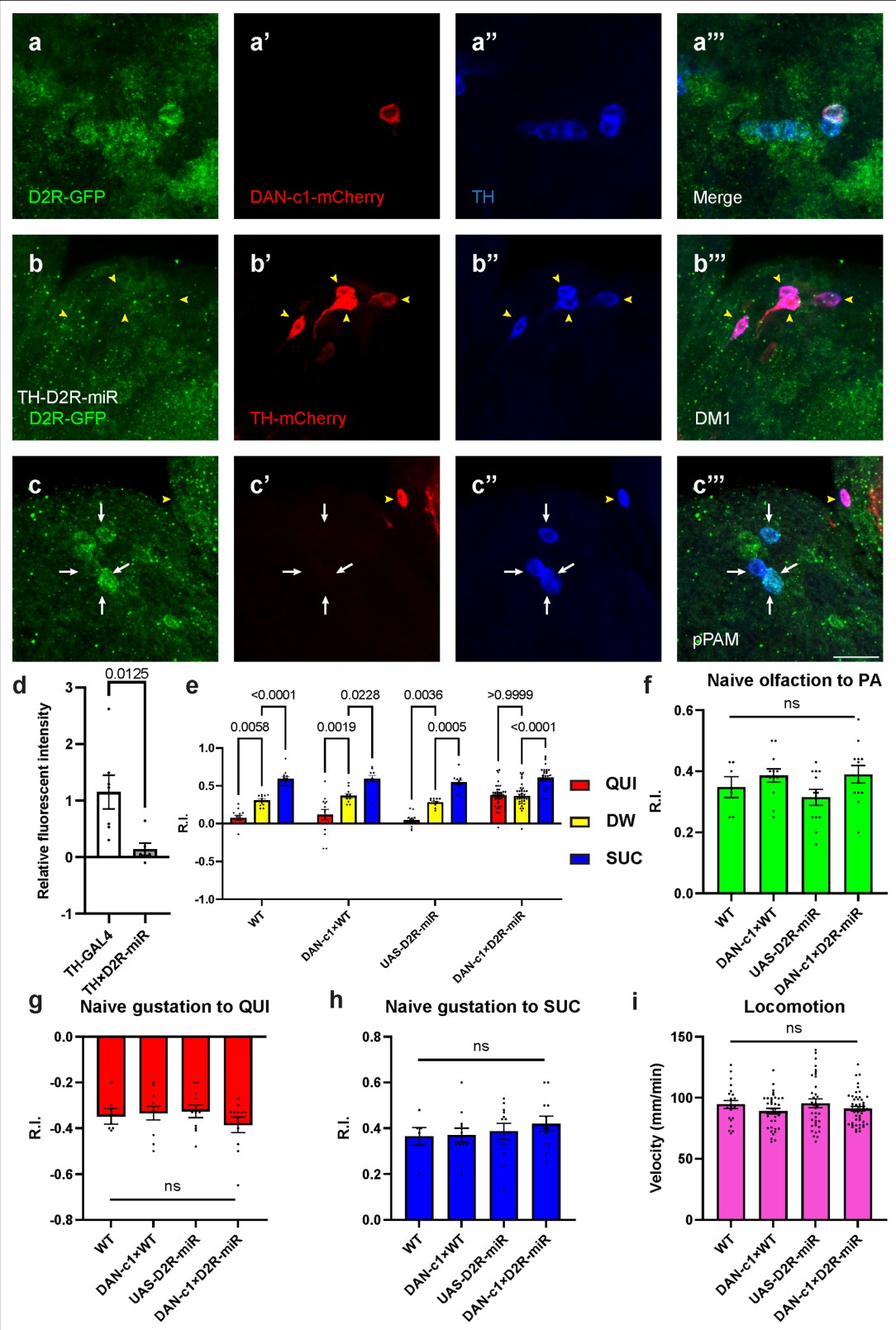

**Figure 4.** Presynaptic D2-like receptor (D2R) in DAN-c1 is necessary for larval aversive learning. (**a**) D2R is expressed in DAN-c1. The expression pattern of D2R is shown with tagged GFP, DAN-c1 is marked by mCherry, and all dopaminergic neurons (DANs) are marked with TH antibody (blue). (**b–c**) Knockdown of D2R by D2R-miR reduces fluorescent intensities of D2R-tagged GFP (D2R-GFP). The TH-GAL4 driver is used to express mCherry. The intensity of D2R-tagged GFP is reduced in the DM1 cluster (yellow arrowheads in b), which is still intact in primary protocerebral anterior medial

*Figure 4 continued on next page*

*Figure 4 continued*

(pPAM) neurons (white arrows in c). (**d**) D2R-GFP fluorescent intensity is quantified by standardizing the values in DM1 with those in the pPAM. Data are shown as mean ± SEM. For the TH-GAL4 group, N=7 brains; for the TH×D2R-miR group, N=6 brains. Unpaired t-test, p=0.0125. (**e**) Knockdown of D2R in DAN-c1 by D2R-miR impairs larval aversive learning. QUI quinine; DW, distilled water; SUC, sucrose. Data are shown as mean ± SEM. Two-way ANOVA, Tukey's multiple comparison test, p<0.0001 for interaction p-values. For N numbers, see *Supplementary file 5*. (**f–i**) D2R knockdown in DAN-c1 does not affect naïve sensory and motor functions. Data are shown as mean ± SEM. One-way ANOVA, Tukey's multiple comparison test. For N numbers, see *Supplementary file 5*. Scale bar: 20 μm. Note: *Figure 4—figure supplement 2* and *Figure 4—figure supplement 3* show additional information on naïve sensory and motor functions in larvae related to D2R-miR experiments. N numbers of immunostaining for each strain can be found in *Supplementary file 2*.

The online version of this article includes the following figure supplement(s) for figure 4:

**Figure supplement 1.** D2-like receptors (D2Rs) in DAN-d1 and DAN-g1 are not necessary for aversive learning.

**Figure supplement 2.** Knockdown D2-like receptor (D2R) in DAN-c1 with an RNAi strain also impairs aversive learning.

**Figure supplement 3.** Naïve sensory and motor functions of *Drosophila* larvae.

was paired with US would it induce the corresponding attraction or repulsion of larvae to the odor. Neither odorant alone, US alone, nor temporal dissociation of odorant and US would induce learning responses. Third, the odor responses are odor specific. When applied to a second odorant that was not used for training, larvae only showed learning responses to the odor paired with US. This result ruled out the explanation of a general olfactory suppression and indicated larvae can discriminate and specifically alter the responses to the odor paired with US. Although the two-odor reciprocal training is not used, these results can show the association of US and the corresponding paired odor. Finally, well-known learning deficit mutants did not show learned responses in this learning paradigm. Honjo et al. tested mutants (e.g. *rut* and *dnc*), which showed learning deficits in the adult stage with two odor reciprocal learning paradigm (*Tempel et al., 1983*; *Dudai et al., 1976*). These mutant larvae also failed to show learning responses when tested with the single odor larval learning paradigm. Combining all the evidence above, we believe this single odor larval learning paradigm is a robust and reliable paradigm for larval associative learning assays, composing essential characteristics of classical conditioning. Previously, we applied this paradigm to investigate the roles of MB serotonin receptors (5-HT7) in larval appetitive learning (*Ganguly et al., 2020*). In this study, we used two distinct odorants (PA and propionic acid), as well as two D2R knockdown strains (UAS-miR and UAS-RNAi for D2R). We obtained similar results for larvae with D2R knockdown in DAN-c1 using different odorants or D2R knockdown strains. In addition, our naïve olfactory, naïve gustatory, and locomotion data ruled out the possibilities that the responses were caused by impaired sensory or motor functions. Comparison with the control group (odor paired with DW) ruled out the potential effects if habituation existed. All these results support this single odor learning paradigm as reliable to assess the learning abilities of *Drosophila* larvae. The failure of reduction in R.I. when larvae with D2R knockdown in DAN-c1 were trained in QUI paired with the odorant is caused by a deficit in aversive learning ability.

## Insights into the neuronal circuits underlying larval olfactory associative learning

MBNs and DANs play important roles in *Drosophila* associative learning in both larvae (*Cognigni et al., 2018*) and adults (*Cognigni et al., 2018*; *Owald and Waddell, 2015*; *Davis, 2005*; *Aso et al., 2014a*). Combining our results with previous learning circuitry research in adult flies and larvae, we hypothesized the mechanism underlying larval olfactory associative learning. Olfactory information (odors, conditioned stimulus [CS]) is received by olfactory sensory neurons, then transmitted to the MB via PNs (*Jefferis et al., 2002*). The MB is a primary learning center of *Drosophila* and composed of Kenyon cells (KCs, or MBNs) (*Heisenberg et al., 1985*; *de Belle and Heisenberg, 1994*; *Davis, 2011*). Their dendrites form the calyx, receiving olfactory information from PNs. The axons converge into peduncles, then branch into the vertical and medial lobes. Distinct gustatory cues (taste, unconditioned stimulus [US]) are sensed by gustatory sensory neurons and transferred to DANs in different clusters (*Figure 7a*). DAN-c1 in the DL1 cluster mediates aversive cues, projects to the LP in the MB. The plasticity of synapses from MBNs to MBONs may be negatively modulated by dopamine, like those in adults (*Owald and Waddell, 2015*; *Séjourné et al., 2011*; *Owald et al., 2015*). The

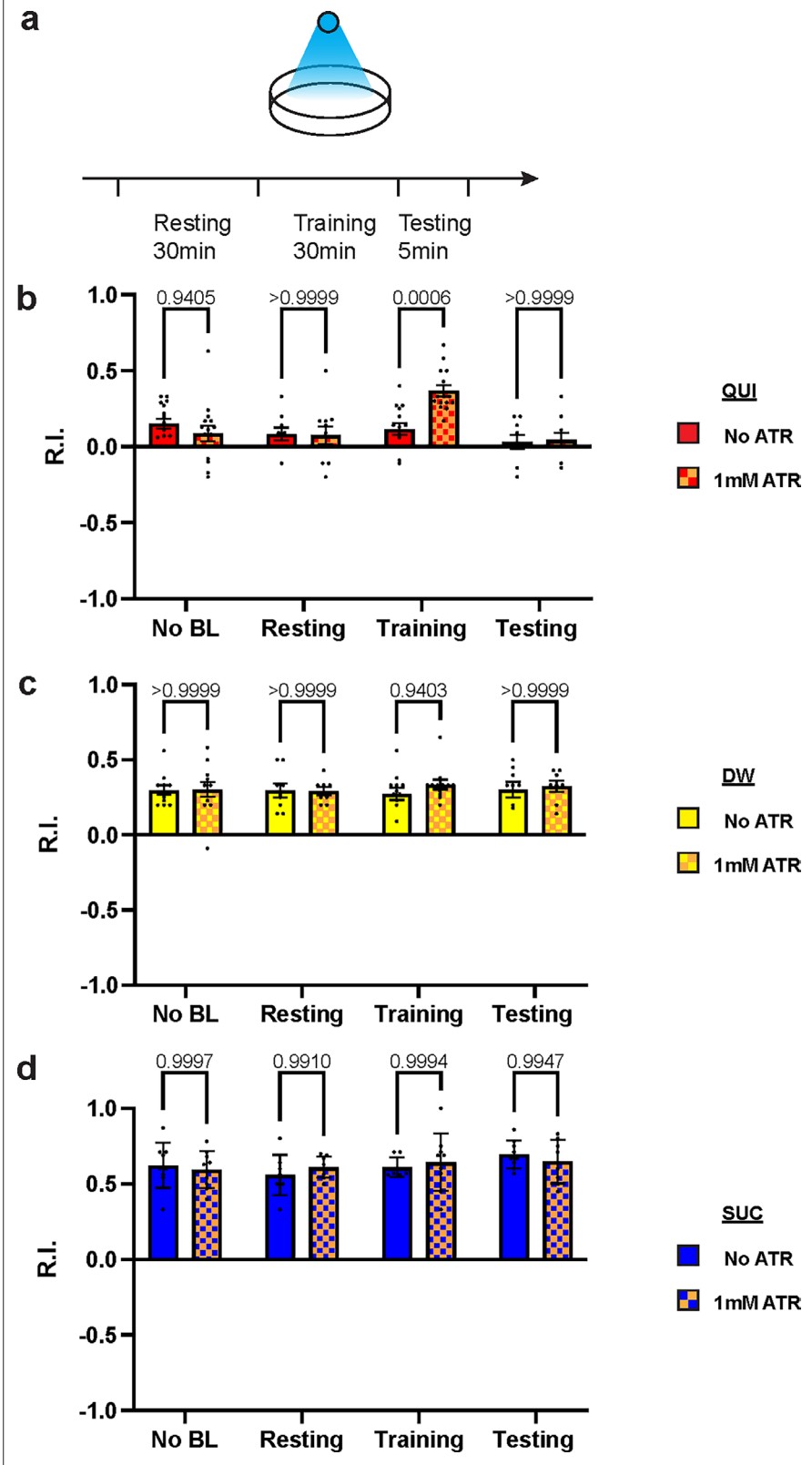

**Figure 5.** Over-excitation of DAN-c1 impairs larval aversive learning. (**a**) A schematic diagram of optogenetic manipulations of neuronal excitability in DAN-c1 during distinct stages in learning. (**b–d**) Activation of DAN-c1 during training impairs larval aversive learning (**b**), while keeping appetitive learning intact (**d**). Unconditioned stimuli used were quinine (QUI) (**b**) and sucrose (SUC) (**d**). No learning behaviors are observed in the control

*Figure 5 continued on next page*

*Figure 5 continued*

distilled water (DW) groups (**c**). Third-instar larvae with ChR2 expression in DAN-c1 are used. *ATR,* all-trans-retinal. Data are shown as mean ± SEM. Two-way ANOVA, Tukey's multiple comparison test. In QUI group (**b**), p=0.0009 for interaction; in DW group (**c**), p=0.8367 for interaction, p=0.9750 for row factor (training stages), and p=0.4872 for column factor (whether with ATR); in SUC group (**d**), p=0.6247 for interaction, p=0.2550 for row factor (training stages), and p=0.9437 for column factor (whether with ATR). For N numbers, see *Supplementary file 6*.

The online version of this article includes the following figure supplement(s) for figure 5:

**Figure supplement 1.** Over-excitation of dopaminergic neurons (DANs) during learning impairs larval aversive learning.

MBN-MBON synapses in the vertical lobe and peduncle are responsible for attraction, while those in the medial lobe are for repulsion.

When only the odorant appears, the subset of KCs representing this odor may be depolarized, inducing calcium influx as in adult flies (*Davis, 2011*). As a balance exists between compartments across the MB lobes, the response to the odor depends on the naïve olfactory circuits from PNs to the lateral horn. In aversive learning, in addition to olfaction-induced calcium influx, gustatory stimuli also lead to DA release from DAN-c1 and subsequently, activation of $G\alpha_s$ in LP. The co-existence of calcium and $G\alpha_s$ activates a $Ca^{2+}$-dependent AC, rutabaga (rut, as in adults; *Zars et al., 2000*; *Tomchik and Davis, 2009*; *Levin et al., 1992*; *Boto et al., 2014*). Rutabaga converges information from both olfaction and gustation, working as the coincidence detector of associative learning. Its downstream signaling inhibits attractive MBN-MBON synapses in the LP, breaking the balance between distinct compartments. As a result, after learning, the larvae will exhibit repulsion when exposed to the odor again. In contrast, DANs in the pPAM convey appetitive cues, which project to compartments in the medial lobe. The co-existence of olfactory and appetitive gustatory stimuli leads to the inhibition of these repulsive MBN-MBON synapses, inducing attraction.

## The conserved role of DAN-c1 in aversive learning throughout *Drosophila* development

Adult *Drosophila* share similar neuronal circuits of learning with larvae (*Saumweber et al., 2018*; *Aso and Rubin, 2016*; *Cognigni et al., 2018*; *Owald and Waddell, 2015*; *Davis, 2005*; *Aso et al., 2014a*). In adult brains, DANs are classified into 13 clusters, named as protocerebral anterior medial (PAM), protocerebral anterior lateral (PAL), protocerebral posterior medial (PPM), protocerebral posterior lateral (PPL), and protocerebral posterior dorsal (PPD) clusters (*Mao and Davis, 2009*). DANs in the PAM cluster innervate the medial lobe, while those in PPL1 project to the vertical lobe (*Aso et al., 2014a*).

R76F02-AD;R55C10-DBD identifies two DANs in adult brains, MB-MP1 in PPL1 and ALT-PLPC in PPL2ab (*Xie et al., 2018*). MB-MP1 is also named PPL1-γ1pedc, innervating both γ1 and the peduncle of the β lobe (*Aso et al., 2014a*). Activation of this neuron induced aversive learning (*Aso et al., 2010*), and activation of its corresponding MBON-γ1pedc>α/β led to approach (*Aso et al., 2014b*). During metamorphosis, DANs in DL1 develop into the PPL1 cluster, DL2a neurons develop into PPL2ab, and those in pPAM will develop into the PAM cluster (*Hartenstein et al., 2017*). This driver strain only identifies DAN-c1 from DL1 in larvae, which innervates the LP of the MB. Interestingly, previous reports revealed that memory can be transferred from larvae to adults (*Tully et al., 1994*), indicating the maintenance of neuronal circuitry architecture during metamorphosis. This evidence supports that DAN-c1 is the corresponding neuron of PPL1-γ1pedc in larvae, which performs similar functions in aversive learning.

## Pre- and postsynaptic D2Rs regulate cAMP in the MB during aversive learning

The molecular mechanisms underlying *Drosophila* learning have not been fully determined. In a traditional view, gustatory cues elevate DA release, which binds to D1 receptors and then activates $G\alpha_s$. The co-existence of $G\alpha_s$ and calcium elicited by olfactory cues activates rutabaga in axons of MBNs. Rutabaga transforms ATP into cAMP, activating PKA signaling pathway. Mutant flies with either insufficient (*rutabaga*) or excessive cAMP (*dunce*) showed aversive learning deficiency, indicating that the

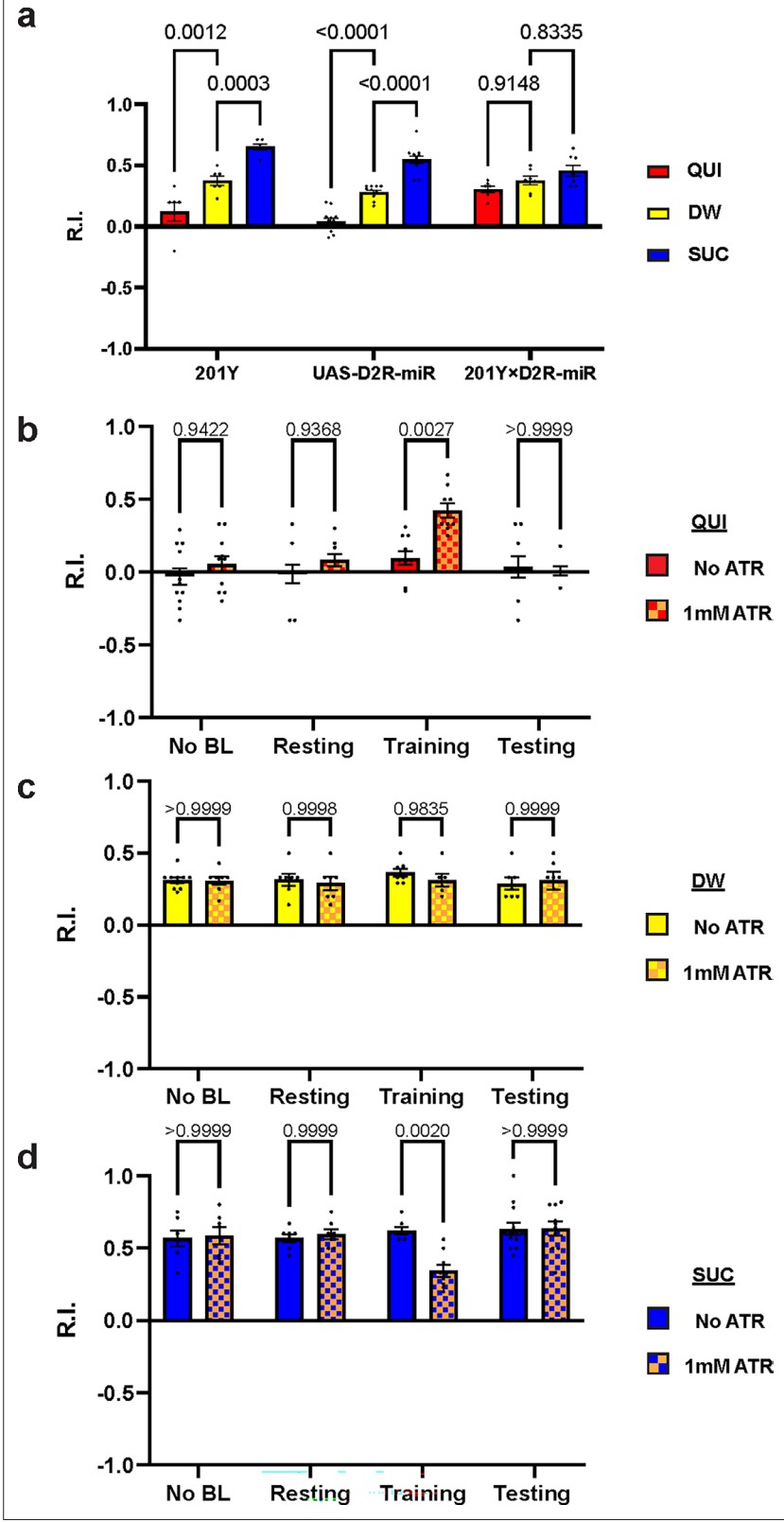

**Figure 6.** D2-like receptor (D2R) in mushroom body is necessary for both aversive and appetitive learning. (**a**) Knockdown of D2R in mushroom body neurons (MBNs) impairs larval aversive and appetitive learning. (**b–d**) Activation of MBNs during training impairs both larval aversive and appetitive learning. Unconditioned stimuli used were quinine (QUI) (**b**) and sucrose (SUC) (**d**). No learning behaviors are observed in the control distilled water

*Figure 6 continued on next page*

*Figure 6 continued*

(DW) groups (**c**). Third-instar larvae with ChR2 expression in MBNs (201Y-GAL4) are used. *ATR,* all-trans retinal. Data are shown as mean ± SEM. Two-way ANOVA, Tukey's multiple comparison test. In D2R knockdown experiments (**a**), p<0.0001 for interaction. In the optogenetic QUI group (**b**), p=0.0259 for interaction; in the DW group (**c**), p=0.8077 for interaction, p=0.7623 for row factor (training stages), and p=0.5846 for column factor (whether with ATR); in the SUC group (**d**), p=0.0035 for interaction. For N numbers, see ***Supplementary file 5 and 6***.

level of cAMP should be kept in an optimal range to achieve aversive learning (***Tempel et al., 1983***; ***Dudai et al., 1976***; ***Figure 7b***).

Our results have shown that D2Rs in DANs and the MB are important for larval aversive learning, suggesting a '*dual brake*' role in regulating cAMP levels in the MBNs through both pre- and post-synaptic components. On the presynaptic side, D2R in DAN-c1 decreases the release of DA under gustatory stimuli, reducing the probability of postsynaptic D1R activation in MBNs. On the postsynaptic side, D2R in MBNs inhibits the coincidence detector rutabaga (AC) both via activation of $G\alpha_{i/o}$ and inhibition of voltage-gated calcium channels (***Neve et al., 2004***), indicating postsynaptic D2R functioning as a '*brake of the coincidence detector*'. Combining these, '*dual brake*' D2R ultimately regulates the MB cAMP level within a physiologically optimal range during aversive learning. In addition, D2R fine-tunes the functional concentration spectrum of DA with higher resolutions, as its DA affinity is 10- to 100-fold greater than D1 receptors. Overall, D2Rs work in a '*dual brake*' system both expanding the representation of a dynamic range of gustatory signal intensity with high signal-to-noise ratio and preventing postsynaptic overexcitation, which increases the reliability of DAN-c1 and MBN circuits for the larval aversive learning (***Figure 7c and d***).

Recent studies showed that the approach/repulsion in adult learning is achieved via inhibition of the repulsive/attractive representing compartments (***Owald and Waddell, 2015***; ***Séjourné et al., 2011***; ***Owald et al., 2015***), which indicates DA inhibits acetylcholine release from MBN to MBON (***Barnstedt et al., 2016***). However, the PKA signaling pathway usually elevates neuronal excitability and increases neurotransmitter release (***Neve et al., 2004***; ***Baik, 2013***), which is contradictory to the recent findings (***Yamada et al., 2024***). In addition to the '*dual brake*' role of D2R, our results suggest a third role of D2R in aversive learning. D1 and D2 receptors can form heteromeric receptors, whose downstream $G\alpha_q$ activates PKC and CaMKII signaling pathways. The activation of these signaling pathways may reduce acetylcholine release (***Wang, 2008***; ***Figure 7c and d***).

## Explanation of the results of thermogenetic and optogenetic experiments

Activation of DAN-c1 with dTRPA1 induced aversive learning, while the repulsion disappeared when DAN-c1 was activated in the QUI group (***Figure 2i***). Our explanation is that QUI stimulation and temperature activation led to over-excitation of DAN-c1, which impaired aversive learning. This is consistent with the learning deficiency in larvae with D2R knockdown in DAN-c1 (***Figure 4e***). Results from optogenetic activation of DAN-c1 during aversive learning also support this (***Figure 5b***). However, in contrast to results with thermo-activation, larvae with optogenetic activation of DAN-c1 showed neither repulsion after being trained with DW (***Figure 5c***), nor did they show reduced attraction in the SUC group (***Figure 5d***). One possible explanation is that the thermo-activation is relatively mild compared to the optogenetic activation. Based on this, thermo-activation of DAN-c1 is still in the physiological range of cAMP under the upper limit (***Figure 7b***), resulting in repulsion in the DW group, neutralized attraction in the SUC group, and impaired repulsion in the QUI group. In contrast, optogenetic activation of DAN-c1 overwhelmed the physiological conditions, leading to failure of repulsion in the DW group (***Figure 5c***). This repulsive failure did not affect appetitive learning (***Figure 5d***), and a stronger over-excitation in the QUI group induced similar failure (***Figure 5b***).

## Distinct DANs may have different roles in larval aversive learning

Although D2Rs are also expressed in DAN-d1 and DAN-g1 (***Figure 4—figure supplement 1d and e***), the knockdown of D2R in these neurons did not impair larval aversive learning (***Figure 4—figure supplement 1f***). For DAN-g1, interestingly, the R.I. from D2R knockdown larvae trained with QUI (aversive learning) showed significant difference when compared to the DW (control) group, but it was also significantly different from the DAN-g1 genetic control group trained with QUI (two-way

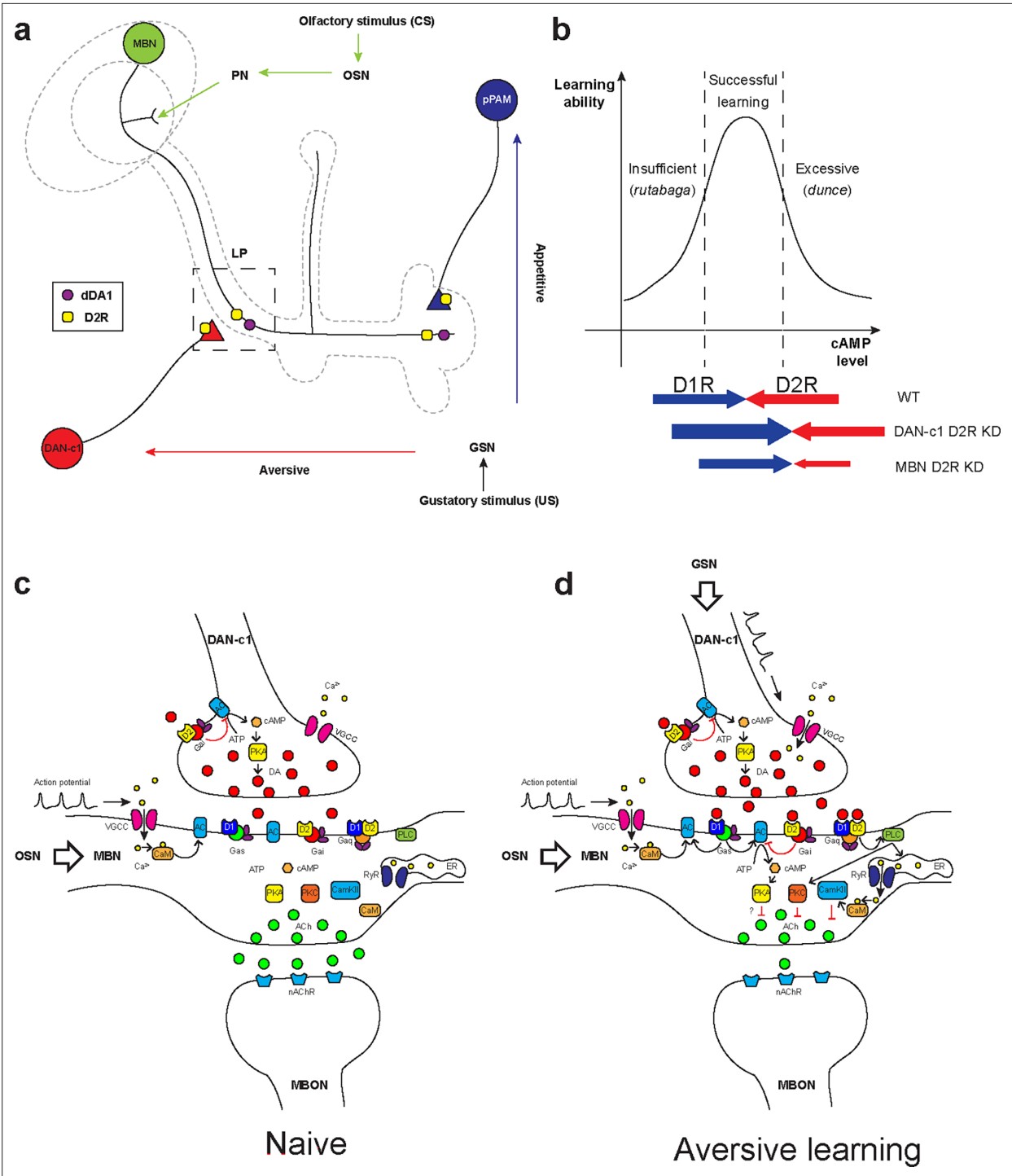

**Figure 7.** Roles of D2-like receptors (D2Rs) in dopaminergic (DANs) and mushroom body (MBNs) neurons during larval olfactory learning. (**a**) A schematic diagram shows the roles of D2R in DANs and MBNs in larval olfactory associative learning. During learning, olfactory stimuli (conditioned stimulus [CS]) are received by olfactory sensory neurons (OSNs) and transmitted to MBNs (green) via projection neurons (PNs). Distinct gustatory stimuli (unconditioned stimulus [US]) are received by gustatory sensory neurons (GSNs) and transferred to different DANs. Aversive stimuli are sent to DAN-c1 (red) in the DL1 cluster which connects the lower peduncle compartment (LP, square in dashed line), while appetitive stimuli are received by pPAM neurons (blue) innervating the medial lobe (ML). D2Rs (yellow square) are expressed in DAN-c1 and pPAM as autoreceptors, regulating dopamine release. Both D2R and dDA1 (magenta circle) are expressed in the MBNs. (**b**) A hypothetical curve showing the relationship between learning ability and cAMP level in the mushroom body. Insufficient cAMP (*rutabaga mutant*) cannot induce learning, while excessive cAMP (*dunce mutant*) also impairs learning. Only the appropriate level of cAMP regulated by the opposing actions of D1R (dDA1) and D2R leads to successful learning in wild-type larvae

*Figure 7 continued on next page*

Figure 7 continued

(WT). Knockdown of D2R in DAN-c1 causes excessive dopamine release, elevating cAMP and resulting in impaired learning. D2R knockdown in MBNs relieves the inhibition effect of D2R, resulting in excessive intracellular cAMP and learning failure. (**c–d**) Potential molecular mechanisms underlying *Drosophila* olfactory learning in the square region as shown in (**a**). (**d**) During aversive learning, olfactory stimuli induce depolarization of MBNs, which activates voltage-gated calcium channels and induces calcium influx. Gustatory stimuli, such as quinine, activate DANs and elevate dopamine (DA) release. D1 receptor (dDA1) activates adenylyl cyclase (AC) and elevates cAMP via G$\alpha_s$, while D2 receptor (D2R) inhibits AC and suppresses cAMP via G$\alpha_{i/o}$. In *Drosophila*, the coincidence detector *rutabaga* (AC) is activated by the existence of both calcium and G$\alpha_s$, converging the olfactory and gustatory stimuli. cAMP activates the PKA signaling pathway, elevating the neuronal excitability. D1 and D2 receptors can also form heteromeric receptors and activate the PLC-PKC and CaMKII signaling pathways via G$\alpha_q$. These pathways inhibit acetylcholine (ACh) release from MBNs to MB output neurons (MBONs), which leads to the avoidance of the learned odor. *Abbreviations:* AC, adenylyl cyclase; ACh, acetylcholine; ATP, adenosine triphosphate; CaM, calmodulin; CaMKII, Ca$^{2+}$/calmodulin-dependent protein kinase II; cAMP, cyclic adenosine monophosphate; CS, conditioned stimulus; DA, dopamine; DANs, dopaminergic neurons; DL, dorsolateral; ER, endoplasmic reticulum; G$\alpha_{i/o}$, G$_{i/o}$ protein $\alpha$ subunit; G$\alpha_q$, G$_q$ protein $\alpha$ subunit; G$\alpha_s$, G$_s$ protein $\alpha$ subunit; GSNs, gustatory sensory neurons; LP, lower peduncle; MBNs, mushroom body neurons; MBONs, mushroom body output neurons; ML, medial lobe; nAChR, nicotinic acetylcholine receptor; OSNs, olfactory sensory neurons; PKA, protein kinase A; PKC, protein kinase C; PLC, phospholipase C; PNs, projection neurons; pPAM, primary protocerebral anterior medial; RyR, ryanodine receptor; US, unconditioned stimulus; VGCC, voltage-gated calcium channel.

ANOVA, Tukey's multiple comparisons, p=0.0002), while not significantly different from the UAS-D2R-miR genetic control group trained with QUI (p=0.2724). Besides, D2R knockdown in DAN-g1 when trained with another odorant propionic acid (ProA) did not show aversive learning deficiency (*Figure 4—figure supplement 2a*). In addition, knockdown of D2R in DAN-g1 using RNAi also did not show aversive learning deficiency when trained with odorant PA (*Figure 4—figure supplement 2b*). This discrepancy may be caused by the different stimulus intensity of distinct odorants, as well as the different efficiency of distinct knockdown methods (microRNA or RNAi strains we used). We suppose that D2Rs in DAN-g1 may partially affect larval aversive learning at a quantitative level but do not play an important role as those in DAN-c1, which will cause a qualitative change when knocked down.

Previous work reported that aversive olfactory learning was induced through the optogenetic activation of DAN-d1, f1, or g1, but not DAN-c1 (*Eschbach et al., 2020*). This discrepancy can be explained as the optogenetic overexcitation of DAN-c1, similar to our optogenetic or D2R knockdown results. Our learning assay results from larvae with D2R knockdown in DAN-d1 or g1 also supported this: aversive learning was not affected by D2R knockdown (*Figure 4—figure supplement 1f*). These data indicate that D2Rs in DAN-d1 or g1 may not be important in larval aversive learning. Additionally, the DAN-c1 strain used in the previous work (SS02160-split-GAL4) not only labels DAN-c1, but also marks other non-DANs, which may affect the results. Besides, live calcium imaging showed that DAN-d1, f1, and g1 responded to the activation of mechanosensory and nociceptive neurons (*Eschbach et al., 2020*), indicating a functional differentiation from gustatory activated DAN-c1 (*Aso et al., 2010*; *Winding et al., 2023*).

In future studies, the molecular signaling downstream of D2R needs to be explored, as well as the comprehensive neuronal circuit architectures of larval learning. The neuronal circuits underlying learning and memory are complex networks, sharing similarities with the regulatory networks of gene expression. Studies of the mechanisms of learning and memory help us understand the essential principles of the nonlinear dynamic characteristics in these complex systems. On one hand, the progress in larval learning provides useful information for helping our understanding of more complex systems, from brains in adult flies to those in mammals. On the other hand, the architecture of larval learning circuits could improve either hardware design or algorithmic architechtures in artificial intelligence, which may bring more powerful tools, such as navigation systems regulating multiple auto-drive vehicles in complex three-dimensional environments.

In conclusion, we explored the expression pattern of D2R in the third-instar larval brain and investigated their roles during larval olfactory learning. D2Rs were found in DAN-c1, and their knockdown induced deficiency in aversive learning. D2Rs were also expressed in MBNs, the knockdown of which impaired both aversive and appetitive learning. This research revealed the important roles of D2Rs in *Drosophila* larval olfactory learning and enriched our understanding regarding the mechanisms underlying the learning process.

## Materials and methods

### Fly stocks

All fly strains used in this study are listed in *Table 1* and *Supplementary file 1*. Flies were maintained on standard medium, which consists of cornmeal, yeast, dextrose, SUC, and agar in water. Flies were kept in a 12/12 hr light/dark cycle at 25°C. Canton-S genotype (WT) and yw[1118] were used as wild-type. Strains carrying more than one transgene were constructed by standard genetic crosses with the w[1118]; CyO/Sco; TM2/TM6 multiple balancer chromosome strain. Strain UAS-Syb::spGFP1-10, LexAop-CD4::GFP11, LexAop-rCD2::RFP /CyO; MB247-LexA::Up16/TM6B was made by chromosome swapping between UAS-Syb::spGFP1-10, LexAop-CD4::GFP11/CyO (II, second chromosome) and LexAop-rCD2::RFP (II).

The D2R gene locates on the X chromosome, with six different alternative splicing products. The GFP-tagged D2R strain is created by inserting a GFP gene into the second intron of D2R gene, generating D2R molecules tagged with GFP (*Nagarkar-Jaiswal et al., 2015a*; *Nagarkar-Jaiswal et al., 2015b*). The D2R-miR strain produces microRNA recognizing the sequence across the third and fourth exons, which is not affected by the GFP insertion (*Figure 4—figure supplement 1c*).

### GRASP

Split-GRASP was used to investigate whether neurons formed synapses (*Macpherson et al., 2015*; *Figure 1—figure supplement 2d*). One part of split GFP tethered to the presynaptic synaptobrevin (UAS-syb::spGFP1-10) was expressed in one type of neuron using UAS/Gal4 binary system, and the complementary split GFP linked to a membrane protein (LexAop-CD4::spGFP11) was expressed in another category of neuron with LexA/LexAop. If synapses between these neurons exist, the split GFPs would form a complete one and be recognized by a mouse antibody (*Figure 1—figure supplement 2a to c*).

### Immunofluorescence

All staining processes were performed in 1.5 mL Eppendorf tubes. Late third-instar (96–100 hr after egg laying) larval brains were dissected in dissection solution (300 mOsmol/L). Brains were fixed in 4% paraformaldehyde (PFA, Electron Microscopy Sciences, Cat. No. 15713) for 1 hr on ice. After three washes (0.1% bovine serum albumin in 10 mM PBS, Sigma Life Sciences A9647), brains were incubated in the blocking and permeabilization solution (0.2% Triton X-100 in 10 mM PBS with 5% normal goat serum; Triton X-100, Sigma, T8532; NGS, Sigma-Aldrich, G9023) for 2 hr at room temperature. Incubation with primary antibodies was done overnight at 4°C. After three washes, brains were incubated in the secondary antibodies for 2 hr. Both the primary and secondary antibodies were diluted in the blocking and permeabilization solution. GFP antibody (Rabbit, Thermo Fisher Scientific, Cat. No. A6455, RRID:AB_221570, 1:1000), TH antibody (Mouse, Immunostar, Cat. No. 22941, RRID:AB_572268, 1:1000), goat anti-rabbit with green fluorescence (Invitrogen, Alexa Fluor 488 conjugate, Cat. No. A-11035, 1:1000), and goat anti-mouse IgG with far-red fluorescence (Alexa Fluor 633 conjugate, Cat. No. 21052, 1:500) secondary antibodies were used. After three washes, brains were transferred and mounted in the Fluoro-Gel with Tris Buffer (Electron Microscopy Sciences, Cat. No. 17985-10) on a piece of micro cover glasses (Electron Microscopy Sciences, Cat. No. 72200-41). Finally, the samples were covered with another piece of micro cover glasses.

A 7-day staining protocol was used for staining GFP-tagged D2R or GRASP, in which brains were fixed in 1% paraformaldehyde in Schneider's insect medium (Sigma Life Sciences, Cat. No. S0146) overnight at 4°C. On the second day, the brains were rinsed and washed twice with PAT3 solution (0.5% Triton X-100 in 10 mM PBS with 0.5% BSA), each for 1 hr. Then brains were incubated in the blocking and permeabilization solution (3% NGS in PAT3) for 2 hr at room temperature. Incubation of primary antibodies was done overnight at 4°C. On the third day, brains were rinsed and washed twice with PBT solution, then incubated in the secondary antibodies for 5 days. Both the primary and secondary antibodies were diluted in the PBTN solution. In the staining of GFP-tagged D2R, GFP antibody (Rabbit, Thermo Fisher Scientific, Cat. No. A6455, RRID:AB_221570, 1: 1000), TH antibody (Mouse, Immunostar, Cat. No. 22941, RRID:AB_572268, 1:1000), and mCherry antibody (Rat, Thermo Fisher Scientific, Cat. No. M11217, RRID:AB_2536611, 1:1000) were used. For staining of GRASP, GFP antibody (Mouse, Sigma-Aldrich, Cat. No. G6539, RRID:AB_259941, 1:100) was used. Goat anti-rabbit with green fluorescence (Invitrogen, Alexa Fluor 488 conjugate, Cat. No. A-11035, 1:1000),

goat anti-mouse with green fluorescence (Alexa Fluor 488 conjugate, Cat. No. A11029, 1:1000), goat anti-mouse with far-red fluorescence (Alexa Fluor 633 conjugate, Cat. No. A21052, 1:500), and goat anti-rat with red fluorescence (Alexa Fluor 546 conjugate, Cat. No. A-11081, 1:1000) secondary antibodies were used. On the seventh day, brains were rinsed and washed twice with PBT solution and then transferred and mounted in the Fluoro-Gel with Tris Buffer on a piece of micro cover glasses. Finally, the samples were covered with another piece of micro cover glasses.

## Confocal imaging

All images were obtained with a Zeiss Laser Scanning Microscope 510 (LSM510, Carl Zeiss, Inc, USA). Under ×40 and ×100 objective magnifications, images were collapsed from confocal stacks of 1.0 µm optical slices. Under ×25 objective magnifications, images were collapsed from confocal stacks of 2.0 µm optical slices. Under ×10 objective magnifications, images were collapsed from confocal stacks of 12.5 µm optical slices. ImageJ software was used to remove other signals outside of the MBs, as the background noise in GRASP is strong.

## Larval olfactory learning assays

All control strains used in learning assays were homozygous except DAN-c1×WT, while all experimental groups (D2R knockdown, thermogenetics, and optogenetics) used were heterozygous by crossing the corresponding control strains. The single odor learning paradigm was slightly modified from previous publications (*Honjo and Furukubo-Tokunaga, 2005*; *Honjo and Furukubo-Tokunaga, 2009*). In brief, 25–50 third-instar larvae (92–96 hr after egg laying) were trained on a 2.5% agar plate (100 mm Petri dish) covered with 2 mL of 1 M SUC solution (Sigma, Cat. No. S1888) or 0.1% QUI hemisulfate solution (Sigma, Cat. No. 22640). DW was used as a control. An odorant PA (10 µL, Sigma-Aldrich, Cat. No. 109584) was placed on a small piece of filter paper (0.25 cm$^2$ square) inside the lid. After 30 min, larvae were rinsed and transferred to the middle line of a new 2.5% agar plate. A small piece of filter paper (0.25 cm$^2$ square) with 2.5 µL PA was placed on one side of the plate, while DW on the other side. After 5 min, the numbers of larvae in the two semicircular areas were counted, and the R.I. was calculated with the following equation (*Figure 2e*):

$$R.I. = \frac{\# \, of \, larvae \, on \, odorant \, side \, in \, dashline - \# \, of \, larvae \, on \, DW \, side \, in \, dashline}{Total \, \# \, of \, larvae \, on \, two \, sides \, in \, dashlines}$$

## Naïve olfactory test

Larvae were transferred into the midline of test plates. 2.5 µL of odorant were added on a piece of filter paper (0.25 cm$^2$ square) on one side and DW on the other side. The number of larvae in two semicircular areas was counted, and the R.I. was calculated after 5 min.

## Naïve gustatory test

A Petri dish with a median separator was used. Both sides were filled with 1% agar, with 2 mL of DW on the control side, and with 1 M SUC solution, or 0.1% QUI hemisulfate solution on the test side. Twenty larvae were put on each side near the midline and allowed to move for 5 min. Gustatory R.I. was calculated using the larvae numbers on two sides (*Lilly and Carlson, 1990*).

## Larval locomotion assay

Individual larvae were placed on the surface of a plate of 2.5% agar mixed with 1 mL India ink. They were allowed to acclimate for 1 min, and then a video was recorded for 30 s using a Moticam3 digital camera (Motic) and Motic Images Plus 2.0 software. The video was analyzed by the MTrack2 plug-in (from http://valelab.ucsf.edu/~nico/IJplugins/MTrack2.html) in ImageJ. The path was recorded; scores were quantified as the length traveled per minute as previously described (*Varga et al., 2014*). As the locomotion speed of DAN-c1 homozygous was slow, DAN-c1×WT was used as the control group.

## Learning assays with thermogenetics

In learning assays with thermogenetics, 25–50 third-instar larvae (92–96 hr after egg laying) were trained on a 2.5% agar plate (100 mm Petri dish) covered with 2 mL of 1 M SUC or 0.1% QUI. DW was used as a control. An odorant PA was placed on a small piece of filter paper (0.25 cm$^2$ square) inside

the lid. Training plates were put in a water bath either under 22°C or 34°C. After 30 min, larvae were rinsed and transferred to the testing plate. After 5 min, the R.I. was calculated.

## Learning assays with optogenetics

In learning assays with optogenetics, egg laying plates with 1 mM all-trans retinal (ATR, Sigma, Cat. No. R2500) were used. ATR is a necessary light-isomerizable chromophore for ChR2, which is not synthesized by *Drosophila* (*Qi et al., 2017*; *Xiao et al., 2014*).

Around 50 third-instar larvae were trained in a 35 mm Petri dish with 2 mL of either 1 M SUC or 0.1% QH solutions. DW was used as a control. During training, an odorant was placed on a small piece of filter paper (0.25 cm$^2$ square) inside the lid. To activate channelrhodopsin2, an LED (Luxeon Rebel Color LEDs, 07040 PB000-D, wavelength 470 nm) with a power supply (GW Instek, Laboratory DC power supply Model GPS-1830D) was used. The intensity of the blue light was 25 mW, measured by a laser power meter (Sanwa, LP1). After being trained for 30 min, larvae were rinsed and transferred to the middle line of a 2.5% agar plate in 100 mm test plate. A small piece of filter paper (0.25 cm$^2$ square) with PA was placed on one side of the plate, while DW on the other side. Then, the number of larvae in the two semicircular areas was counted, and the R.I. was calculated after 5 min.

## Quantification of D2R knockdown

Quantification of the fluorescent intensity of D2R knockdown was performed as follows. TH signals were used to define DANs, and the mean fluorescent intensity of GFP in each neuron was calculated with subtraction of the background. The mean intensity of DM1 was divided by that of pPAM in each brain, and the value in the knockdown group was subsequently normalized with the control group.

## Statistical analysis

Information for statistical analysis is provided in figure legends, which was conducted by Prism 10 (GraphPad Software, LLC). In brief, two-way ANOVA and Tukey's multiple comparison test were used in *Figures 2f–j and 4e*, *Figures 5b–d and 6a–d*, and *Figure 5—figure supplement 1*; unpaired t-test was used in *Figure 4d*; one-way ANOVA and Tukey's multiple comparison test were used in *Figure 4f–i*; two-way ANOVA and Dunnett's multiple comparison test were used in *Figure 4—figure supplement 1*, and *Figure 4—figure supplement 2a and b*; one-way ANOVA and Dunnett's multiple comparison test were used in *Figure 4—figure supplement 3a to i*.

# Acknowledgements

This work was partially supported by an NIH grant (AG065925) and an International Collaboration Grant from Korea Institute of Science & Technology (Brain Science Institute), Seoul, Korea. CQ was a recipient of the SEA and GSR awards from Ohio University. We thank Dr. J Hirsh (University of Virginia), Dr. M Wu (Johns Hopkins University), Dr. M Zlatic and Dr. C Eschbach (HHMI Janelia Research Campus), Dr. M Gallio (Northwestern University), Dr. A Kopin (Tufts-New England Medical Center), Dr. S Tanda (Ohio University), Dr. B Condron (University of Virginia), Dr. T Kitamoto (University of Iowa) for their kind gift of fly strains.

# Additional information

### Funding

| Funder | Grant reference number | Author |
| --- | --- | --- |
| National Institutes of Health | AG065925 | Daewoo Lee |
| Korea Institute of Science & Technology (Brain Science Institute) | | Daewoo Lee |

The funders had no role in study design, data collection and interpretation, or the decision to submit the work for publication.

## Author contributions
Cheng Qi, Conceptualization, Formal analysis, Investigation, Methodology, Writing – original draft, Writing – review and editing; Cheng Qian, Emma Steijvers, Robert A Colvin, Writing – review and editing; Daewoo Lee, Conceptualization, Supervision, Funding acquisition, Writing – review and editing

## Author ORCIDs
Cheng Qi ⓘ https://orcid.org/0000-0002-5716-5020
Daewoo Lee ⓘ https://orcid.org/0009-0004-1565-3204

Reviewer #1 (Public review): https://doi.org/10.7554/eLife.100890.3.sa1
Reviewer #2 (Public review): https://doi.org/10.7554/eLife.100890.3.sa2
Author response https://doi.org/10.7554/eLife.100890.3.sa3

# Additional files

## Supplementary files
Supplementary file 1. Other strains used in this study.

Supplementary file 2. N numbers for brain samples used in Figures.

Supplementary file 3. Summary of R76F02AD;R55C10DBD (DAN-c1) identifying patterns in larval brains. For the strain R76F02AD; R55C10DBD, 22 third-instar larval brains expressing GFP or SytGFP and DenMark were examined, and all of them clearly identified DAN-c1. Half of them only identified DAN-c1, the rest had 1–5 weakly identified cells without neurites, and barely 1 or 2 strongly identified cells appeared. These non-DAN-c1 neurons were seldom dopaminergic neurons. In the ventral nerve cord (VNC), 8 out of 12 did not have any identified cells, 3 had 2–4 strong identified cells. These data supported that R76F02AD;R55C10DBD exclusively labels DAN-c1 in third-instar larval brains.

Supplementary file 4. N numbers and p-values for learning assays with thermogenetics.

Supplementary file 5. N numbers for D2-like receptor (D2R) knockdown experiments.

Supplementary file 6. N numbers for learning assays with optogenetics.

MDAR checklist

## Data availability
Data generated and analyzed during this study can be found at https://doi.org/10.5061/dryad.kwh70rzhq.

The following dataset was generated:

| Author(s) | Year | Dataset title | Dataset URL | Database and Identifier |
|---|---|---|---|---|
| Qi C, Qian C, Steijvers E, Colvin R, Lee D | 2025 | A pair of dopaminergic neurons DAN-c1 mediate Drosophila larval aversive olfactory learning through D2-like receptors | https://doi.org/10.5061/dryad.kwh70rzhq | Dryad Digital Repository, 10.5061/dryad.kwh70rzhq |

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
