## [Editor Report · eLife Assessment]

This study presents **valuable** findings on the role of dopamine receptor D2R in dopaminergic neurons DAN-c1 and mushroom body neurons (Y201-GAL4 pattern) on aversive and appetitive conditioning. The evidence supporting the claims of the authors is **solid** in the context of their behavioural paradigm. Controls using a reciprocal training protocol would have broadened the scope of their conclusions. The work will be of interest to researchers studying the role of dopamine during learning and memory.

---

## [Referee Report · Reviewer #1 (Public review)]

Summary:

Both flies and mammals have D1-like and D2-like dopamine receptors, yet the role of D2-like receptors in Drosophila learning and memory remains underexplored. This paper investigates the role of the D2-like dopamine receptor D2R in single pairs of dopaminergic neurons (DANs) during single-odor aversive learning in the Drosophila larva. First, confocal imaging is used to screen GAL4 driver strains that drive GFP expression in just single pairs of dopaminergic neurons. Next, thermogenetic manipulations of one pair of DANs (DAN-c1) suggest that DAN-c1 activity during larval aversive learning is important. Confocal imaging is then used to reveal expression of D2R in the DANs and mushroom body of the larval brain. Finally, optogenetic activation during training phenocopies D2R knockdown in these neurons: aversive learning is impaired when DAN-c1 is targeted, while appetitive and aversive learning are impaired when the mushroom body is manipulated. Finally, a model is proposed in which D2R limits excessive dopamine release to facilitate successful olfactory learning.

Strengths:

The paper convincingly reproduces prior findings that demonstrated D2R knockdown in DL1 DANs or the mushroom body impairs aversive olfactory learning in Drosophila larvae (Qi and Lee, 2014; doi:10.3390/biology3040831). These previous findings were built upon and extended with a comprehensive confocal imaging screen of 57 GAL4 drivers that identified tools driving GFP expression in individual DANs. One of the drivers, R76F02-AD; R55C10-DBD, was consistently shown to label DAN-c1 neurons and no other DANs in the larval brain. Confocal imaging is also used to demonstrate that GFP-tagged D2R is expressed in most DANs and the mushroom body. Behavioral experiments demonstrate that driving D2R knockdown in DAN-c1 neurons impairs aversive learning, as do other loss-of-function manipulations of DAN-c1 neurons.

Limitations:

(1) The single-odor paradigm used to train larvae does not have the advantages of a more conventional balanced or reciprocal training paradigm. The paper describes how the single-odor experimental design could be controlled for non-associative effects, but does not provide an independent validation of the control experiments performed by a different research group using different odors and genotypes 15 to 20 years ago (see Honjo and Furukubo-Tokunaga, 2005; doi:10.1523/jneurosci.2135-05.2005 and Honjo and Furukubo-Tokunaga, 2009; doi:10.1523/jneurosci.1315-08.2009). Whether the involvement of DAN-c1 for aversive learning generalizes to standard paradigms remains unclear (see Eschbach et al., 2020; doi:10.1038/s41593-020-0607-9 and Weber et al., 2023; doi:10.7554/elife.91387.1).

(2) In 11 of 22 larval brains examined in the paper, R76F02-AD; R55C10-DBD appears to drive GFP expression in 1 to 8 additional non-dopaminergic neurons (Figure S1P and Table S3). Of the remaining 11 brains, 4 of their corresponding ventral nerve cords also have expression in 2 to 4 neurons (Table S3). Therefore, experiments involving with the R76F02-AD; R55C10-DBD driver could be manipulating the activity of additional neurons in around 60% of larvae. The conclusions of the paper would be strengthened if key experiments were repeated with other GAL4 drivers that may label DAN-c1 with even greater specificity, such as SS03066 (Truman et al., 2023; doi:10.7554/elife.80594) or MB320C (Hige et al., 2015; doi:10.1016/j.neuron.2015.11.003).

(3) Successful immunostaining with an anti-D2R antibody (Draper et al., 2007; doi:10.1002/dneu.20355 and Love et al., 2023; doi:10.1111/gbb.12836) could validate GFP-tagged D2R expression (Figure 3) in the same way that TH immunostaining was used throughout the paper to determine whether neurons were dopaminergic.

(4) The paper proposes a model in which DAN-c1 activity conveys an aversive teaching signal (Figure 2f) but excessive artificial DAN-c1 activation causes excessive dopamine release that impairs aversive learning (Figures 2i and 5b). According to this model, thermogenetic DAN-c1 activation during training with water or sucrose conveys an aversive teaching signal that reduces performance (Figure 2i) whereas optogenetic DAN-c1 activation does not due to excessive dopamine release (Figures 5c and 5d). The model suggests that optogenetic DAN-c1 activation is strong enough to cause excessive dopamine release by itself whereas thermogenetic DAN-c1 activation can only achieve the same outcome when it occurs in conjunction with natural DAN-c1 activation evoked by quinine. Therefore, an experiment with weaker optogenetic DAN-c1 activation (with lower intensity light or pulsed at a lower frequency) during water or sucrose training would be expected to convey an aversive teaching signal rather than excessive dopamine release, reducing performance. Such an experiment could reconcile the differing thermogenetic and optogenetic results of the paper.

---

## [Referee Report · Reviewer #2 (Public review)]

Summary:

The study wanted to functionally identify individual DANs that mediate larval olfactory learning. Then search for DAN-specific driver strains that mark single dopaminergic neurons, which subsequently can be used to target genetic manipulations of those neurons. 56 GAL4 drivers identifying dopaminergic neurons were found (Table 1) and three of them drive the expression of GFP to a single dopaminergic neuron in the third-instar larval brain hemisphere. The DAN driver R76F02-AD;R55C10-DBD appears to drive the expression to a dopaminergic neuron innervating the lower peduncle (LP), which would be DAN-c1.

Split-GFP reconstitution across synaptic partners (GRASP) technique was used to investigate the "direct" synaptic connections from DANs to the mushroom body. Potential synaptic contact between DAN-c1 and MB neurons (at the lower peduncle) were detected.

Then single odor associative learning was performed and thermogenetic tools were used (Shi-ts1 and TrpA1). When trained at 34{degree sign}C, the complete inactivation of dopamine release from DAN-c1 with Shibirets1 impaired aversive learning (Figure 2h), while Shibirets1 did not affect learning when trained at room temperature (22 {degree sign}C). When paired with a gustatory stimulus (QUI or SUC), activation of DAN-c1 during training impairs both aversive and appetitive learning (Figure 2k).

Then examined the expression pattern of D2R in fly brains and were found in dopaminergic neurons and the mushroom body (Figure 3). To inspect whether the pattern of GFP signals indeed reflected the expression of D2R, three D2R enhancer driver strains (R72C04, R72C08, and R72D03-GAL4) were crossed with the GFP-tagged D2R strain.

D2R knockdown (UAS-RNAi) in dopaminergic neurons driven by TH-GAL4 impaired larval aversive learning. Using a microRNA strain (UAS-D2R-miR), a similar deficit was observed. Crossing the GFP-tagged D2R strain with a DAN-c1-mCherry strain demonstrated the expression of D2R in DAN-c1 (Figure 4a). Knockdown of D2R in DAN-c1 impaired aversive learning with the odorant pentyl acetate, while appetitive learning was unaffected (Figure 4e). Sensory and motor functions appear not affected by D2R suppression.

To exclude possible chronic effects of D2R knockdown during development, optogenetics was applied at distinct stages of the learning protocol. ChR2 was expressed in DAN-c1, and blue light was applied at distinct stages of the learning protocol. Optogenetic activation of DAN-c1 during training impaired aversive learning, not appetitive learning (Figure 5b-d).

Knockdown of D2Rs in MB neurons by D2R-miR impaired both appetitive and aversive learning (Figure 6a). Activation of MBNs during training impairs both larval aversive and appetitive learning.

Finally, based on the data the authors propose a model where the effective learning requires a balanced level of activity between D1R and D2R (Figure 7).

Strengths:

The work is well written, clear, and concise. They use well documented strategies to examine GAL4 drivers with expression in a single DAN, behavioral performance in larvae with distinct genetic tools including those to do thermo and optogenetics in behaving flies. Altogether, the study was able to expand our understanding of the role of D2R in DAN-c1 and MB neurons in the larva brain.

The study successfully examined the role of D2R in DAN-c1 and MB neurons in olfactory conditioning. The conclusions are well supported by the data and the model of adequate levels of cAMP (Figure 7b) appears to be able to explain a poor memory after insufficient or excessive cAMP signaling. The study provides insight into the role of D2R in associative learning expanding our understanding and might be a reference similarly to previous key findings (Qi and Lee, 2014, https://doi.org/10.3390/biology3040831).

---

## [Author Response]

The following is the authors’ response to the original reviews

**Public Reviews:**

**Reviewer #1 (Public review):**
Weakness#1: The authors claim to have identified drivers that label single DANs in Figure 1, but their confocal images in Figure S1 suggest that many of those drivers label additional neurons in the larval brain. It is also not clear why only some of the 57 drivers are displayed in Figure S1.

As described in the Results section, we screened 57 GAL4 driver lines based on previous reports. These included drivers that had been shown to label a single dopaminergic neuron (DAN) or a small subset of DANs in the larval or adult brain hemisphere, suggesting potential for specific DAN labeling in larvae.

In Figure 1, TH-GAL4 was used to cover all neurons in the DL1 cluster, while R58E02 and R30G08 were well known drivers for pPAM. Fly strains in Figure 1h, k, l, and m were reported as single DAN strains in larvae[1], while strains in Figure 1e, f, g were reported identifying only several DANs in adult brains[2,3]. We examined these strains and only some of them labeled single DANs in 3rd instar larval brain hemisphere (Figure 1f, g, h, l and m). Among them, only strains in Figure 1f and h labeled single DAN in the brain hemisphere, without labeling other non-DANs. Other strains labeled non-DANs in addition to single DANs (Figure 1g, l and m). Taking ventral nerve cord (VNC) into consideration, strain in Figure 1h also labeled neurons in VNC (Figure S1e), while strain in Figure 1f did not (Figure S1c).

In summary, the driver shown in Figure 1f (R76F02AD;R55C10DBD, labeling DAN-c1) is the only line we identified that labels a single DAN in the 3rd instar larval brain hemisphere without additional labeling. The other lines shown in Figure 1 (g, h, l, m) label a single DAN but also include some non-DANs. Figure 1 focuses on strains that label a single or a pair of DANs.

Labeling patterns for all 57 driver lines are summarized in Table 1. Figure S1 includes representative examples; full confocal images for all screened strains are available upon request, as stated in the figure legend.

Weakness #2: Critically, R76F02-AD; R55C10-DBD labels more than one neuron per hemisphere in Figure S1c, and the authors cite Xie et al. (2018) to note that this driver labels two DANs in adult brains. Therefore, the authors cannot argue that the experiments throughout their paper using this driver exclusively target DAN-c1.

Figure S1c shows a single dopaminergic (DA) neuron in each brain hemisphere. While additional GFP-positive signals were occasionally observed, they did not originate from the cell bodies of DA neurons, as these were not labeled by the tyrosine hydroxylase (TH) antibody. These additional GFP signals primarily appeared to be neurites, including axonal terminals, although we cannot rule out the possibility that some represent false-positive signals or weakly stained non-neuronal cell bodies. This interpretation is based on the analysis of 22 third-instar larval brains.

To clarify this point in the manuscript, we added the following sentence to the Results section: “Based on the analysis of 22 brain samples, we observed this driver strain labels one neuron per hemisphere in the third-instar larval brain (Figure 2a–d, Figure S1c, Table S3).” Additionally, Table S3 was included to summarize the DAN-c1 labeling pattern across all 22 samples. An enlarged inset highlighting GFP-positive signals was also added to Figure S1c.

Weakness #3: Missing from the screen of 57 drivers is the driver MB320C, which typically labels only PPL1-γ1pedc in the adult and should label DAN-c1 in the larva. If MB320C labels DAN-c1 exclusively in the larva, then the authors should repeat their key experiments with MB320C to provide more evidence for DAN-c1 involvement specifically.

We thank the reviewer for this insightful suggestion. The MB320C driver primarily labels the PPL1-γ1pedc neuron in the adult brain, along with one or two additional weakly labeled cells. It would indeed be interesting to examine the expression pattern of this driver in third-instar larval brains. If it is found to label only DAN-c1 at this stage, we could consider using it to knock down D2R and assess whether this recapitulates our current findings.

While we agree that this is a promising direction for future studies, we believe it is not essential for the current manuscript, given the specificity of the DAN-c1 driver (please see our response to Reviewer #3 for details). Nonetheless, we appreciate the reviewer’s suggestion, and we recognize that MB320C could be a valuable tool for future experiments.

Weakness #4: The authors claim that the SS02160 driver used by Eschbach et al. (2020) labels other neurons in addition to DAN-c1. Could the authors use confocal imaging to show how many other neurons SS02160 labels? Given that both Eschbach et al. and Weber et al. (2023) found no evidence that DAN-c1 plays a role in larval aversive learning, it would be informative to see how SS02160 expression compares with the driver the authors use to label DAN-c1.

We did not have our own images showing DANs in brains of SS02160 driver cross line. However, Extended Data Figure 1 in the paper of Eschbach et al. shows strongly labeled four neurons on each brain hemisphere[4], indicating that this driver is not a strain only labeling one neuron, DAN-c1.

Weakness #5: The claim that DAN-c1 is both necessary and sufficient in larval aversive learning should be reworded. Such a claim would logically exclude any other neuron or even the training stimuli from being involved in aversive learning (see Yoshihara and Yoshihara (2018) for a detailed discussion of the logic), which is presumably not what the authors intended because they describe the possible roles of other DANs during aversive learning in the discussion.

We agree with the reviewer that the terms “necessary” and “sufficient” may be too exclusive and could unintentionally exclude contributions from other neurons. As noted in the Discussion section, we acknowledge that additional dopaminergic neurons may also play roles in larval aversive learning. To reflect this, we have revised our wording to use “important” and “mediates” instead of the more definitive terms “necessary” and “sufficient,” making our conclusions more accurate and appropriately measured.

Weakness #6: Moreover, if DAN-c1 artificial activation conveyed an aversive teaching signal irrespective of the gustatory stimulus, then it should not impair aversive learning after quinine training (Figure 2k). While the authors interpret Figure 2k (and Figure 5) to indicate that artificial activation causes excessive DAN-c1 dopamine release, an alternative explanation is that artificial activation compromises aversive learning by overriding DAN-c1 activity that could be evoked by quinine.

This is an excellent point, and we agree that we cannot rule out the possibility that artificial activation interferes with aversive learning by overriding the natural activity of DAN-c1 that would normally be evoked by quinine. The observed results with TRPA1 could potentially be attributed to dopamine depletion, inactivation due to prolonged depolarization, or neural adaptation. However, we believe that our hypothesis - that over-excitation of DAN-c1 impairs learning - is more consistent with our experimental findings and with previously published data. Our rationale is as follows: (1) Associative learning in larvae occurs only when the conditioned stimulus (CS, e.g., an odor such as pentyl acetate) and unconditioned stimulus (US, e.g., quinine) are paired. In wild-type larvae, the CS depolarizes a subset of Kenyon cells in the mushroom body (MB), while the US induces dopamine (DA) release from DAN-c1 into the lower peduncle (LP) compartment (Figure 7a). When both stimuli coincide, calcium influx from CS activation and Gαs signaling via D1-type dopamine receptors activate the MB-specific adenylyl cyclase, *rutabaga*, which functions as a coincidence detector (Figure 7d). (2) *Rutabaga* converts ATP to cAMP, activating the PKA signaling pathway and modifying synaptic strength between Kenyon cells and mushroom body output neurons (MBONs) (Figure 7d). These changes in synaptic strength underlie learned behavioral responses to future presentations of the same odor. (3) Our results show that D2R is expressed in DAN-c1, and that D2R knockdown impairs aversive learning. Since D2Rs typically inhibit neuronal excitability and reduce cAMP levels[5], we hypothesize that D2R acts as an autoreceptor in DAN-c1 to restrict DA release. When D2R is knocked down, this inhibition is lifted, leading to increased DA release in response to the US (quinine). The resulting excess DA, in combination with CS-induced calcium influx, would elevate cAMP levels in Kenyon cells excessively - disrupting normal learning processes (Figure 7b). This is supported by studies showing that *dunce* mutants, which have elevated cAMP levels, also exhibit aversive learning deficits[6]. (4) The TRPA1 activation results are consistent with our over-excitation model. When DAN-c1 was artificially activated at 34°C in the distilled water group, this mimicked the natural activation by quinine, producing an aversive learning response toward the odor (Figure 2k or new Figure 2i, DW group). Similarly, in the sucrose group, artificial activation mimicked quinine, producing a learning response that reflected both appetitive and aversive conditioning (Figure 2k, SUC group). (5) Over-excitation impairs learning in the quinine group. When DAN-c1 was activated during quinine exposure, both artificial and natural activation combined to produce excessive DA release. This over-excitation likely disrupted the cAMP balance in Kenyon cells, impairing learning and resulting in failure of aversive memory formation (Figure 2k, QUI group). This phenotype closely mirrors the effect of D2R knockdown in DAN-c1. (6) Optogenetic activation of DAN-c1 during aversive training similarly produced elevated DA levels due to both natural and artificial stimulation. This again would result in MBN over-excitation and a corresponding learning deficit. When optogenetic activation occurred during non-training phases (resting or testing), no additional DA was released during training, and aversive learning remained intact (Figure 5b). (7) Notably, when optogenetic activation was applied during training, we observed no aversive learning in the distilled water group and no reduction in the sucrose group (Figure 5c, 5d). We interpret this as evidence that the optogenetic stimulation was strong enough to cause elevated DA release in both groups, impairing learning in a manner similar to D2R knockdown or TRPA1 overactivation. (8) We extended this over-excitation framework to directly activate Kenyon cells (MBNs). Since MBNs are involved in both appetitive and aversive learning, their over-excitation disrupted both types of learning (Figure 6), further supporting our hypothesis. In summary, we propose that DAN-c1 activity is tightly regulated by D2R autoreceptors to ensure appropriate levels of dopamine release during aversive learning. Disruption of this regulation - either through D2R knockdown or artificial overactivation of DAN-c1 - results in excessive DA release, over-excitation of Kenyon cells, and impaired learning. This over-excitation model is consistent with both our experimental results and prior literature.

Weakness #7: The authors should not necessarily expect that D2R enhancer driver strains would reflect D2R endogenous expression, since it is known that TH-GAL4 does not label p(PAM) dopaminergic neurons.

Just like the example of TH-GAL4, it is possible that the D2R driver strains may partially reflect the expression pattern of endogenous D2R in larval brains. When we crossed the D2R driver strains with the GFP-tagged D2R strain, however, we observed co-localization in DM1 and DL2b dopaminergic neurons, as well as in mushroom body neurons (Figure S3c to h). In addition, D2R knockdown with D2R-miR directly supported that the GFP-tagged D2R strain reflected the expression pattern of endogenous D2R (Figure 4b to d, signals were reduced in DM1). In summary, we think the D2R driver strains supported the expression pattern we observed from the GFP-tagged D2R strain, especially in DM1 DANs.

Weakness #8: Their observations of GFP-tagged D2R expression could be strengthened with an anti-D2R antibody such as that used by Lam et al., (1999) or Love et al., (2023).

Love et al. (2023) used the antibody originally described by Draper et al.[6]. We attempted to use the same antibody in our experiments; however, we were unable to detect clear signals following staining. This may be due to a lack of specificity for neurons in the Drosophila larval brain or incompatibility with our staining protocol. Unfortunately, we were unable to locate a copy of the Lam (1999) paper for further reference.

Weakness #9: Finally, the authors could consider the possibility other DANs may also mediate aversive learning via D2R. Knockdown of D2R in DAN-g1 appears to cause a defect in aversive quinine learning compared with its genetic control (Figure S4e). It is unclear why the same genetic control has unexpectedly poor aversive quinine learning after training with propionic acid (Figure S5a). The authors could comment on why RNAi knockdown of D2R in DAN-g1 does not similarly impair aversive quinine learning (Figure S5b).

We re-analyzed the data related to DAN-g1. Interestingly, knockdown of D2R in DAN-g1 larvae trained with quinine (QUI) showed a significant difference in response index (R.I.) compared to the distilled water (DW) control group. However, it also differed significantly from the DAN-g1 genetic control group trained with QUI (two-way ANOVA with Tukey’s multiple comparisons, *p* = 0.0002), while it was not significantly different from the UAS-D2R-miR genetic control group (*p* = 0.2724). Furthermore, knockdown of D2R in DAN-g1 did not lead to aversive learning deficits when larvae were trained with a different odorant, propionic acid (ProA; Figure S5a). Similarly, using an RNAi line to knock down D2R in DAN-g1 did not result in learning impairment when larvae were trained with pentyl acetate (PA; Figure S5b). These inconsistencies may stem from differences in stimulus intensity across odorants, as well as the variable efficiency of the knockdown strategies (microRNA vs. RNAi). Based on these results, we propose that D2Rs in DAN-g1 may modulate larval aversive learning in a quantitative manner but do not play as critical a role as those in DAN-c1, where knockdown produces a clear qualitative effect. We have added this paragraph to the Discussion section of the manuscript.

**Reviewer #2 (Public review):**
Weakness#1: Is not completely clear how the system DAN-c1, MB neurons and Behavioral performance work. We can be quite sure that DAN-c1;Shits1 were reducing dopamine release and impairing aversive memory (Figure 2h). Similarly, DAN-c1;ChR2 were increasing dopamine release and also impaired aversive memory (Figure 5b). However, is not clear what is happening with DAN-c1;TrpA1 (Figure 2K). In this case the thermos-induction appears to impair the behavioral performance of all three conditions (QUI, DW and SUC) and the behavior is quite distinct from the increase and decrease of dopamine tone (Figure 2h and 5b).The study successfully examined the role of D2R in DAN-c1 and MB neurons in olfactory conditioning. The conclusions are well supported by the data, with the exception of the claim that dopamine release from DAN-c1 is sufficient for aversive learning in the absence of unconditional stimulus (Figure 2K). Alternatively, the authors need to provide a better explanation of this point.

Please refer to our response to Weakness #6 of Reviewer #1 above.

**Reviewer #3 (Public review):**
Weakness #1: It is a strength of the paper that it analyses the function of dopamine neurons (DANs) at the level of single, identified neurons, and uses tools to address specific dopamine receptors (DopRs), exploiting the unique experimental possibilities available in larval Drosophila as a model system. Indeed, the result of their screening for transgenic drivers covering single or small groups of DANs and their histological characterization provides the community with a very valuable resource. In particular the transgenic driver to cover the DANc1 neuron might turn out useful. However, I wonder in which fraction of the preparations an expression pattern as in Figure 1f/ S1c is observed, and how many preparations the authors have analyzed. Also, given the function of DANs throughout the body, in addition to the expression pattern in the mushroom body region (Figure 1f) and in the central nervous system (Figure S1c) maybe attempts can be made to assess expression from this driver throughout the larval body (same for Dop2R distribution).

We thank the reviewer for the positive comments and thoughtful suggestions.

Regarding the R76F02AD; R55C10DBD strain, we examined 22 third instar larval brains expressing GFP, Syt-GFP, or Den-mCherry. All brains clearly labeled DAN-c1. In approximately half of the samples, only DAN-c1 was labeled. In the remaining samples, 1 to 5 additional weakly labeled soma were observed, typically without associated neurites. Only 1 or 2 strongly labeled non-DAN-c1 cells were occasionally detected. These additional labeled neurons were rarely dopaminergic. In the ventral nerve cord (VNC), 8 out of 12 samples showed no labeled cells. The remaining 4 samples had 2–4 strongly labeled cells. These results support our conclusion that the R76F02AD; R55C10DBD combination predominantly and specifically labels DAN-c1 in the third instar larval brain. As for the reviewer’s question about the expression pattern of R76F02AD; R55C10DBD and D2R in the larval body, we agree that this is a very interesting avenue for further investigation. However, our current study is focused on the central nervous system and larval learning behaviors. We hope to explore this question more fully in future work.

We added the following sentence to the Results section: “Based on analysis of 22 brain samples, we believe this driver strain consistently labels one neuron per hemisphere in the third-instar larval brain (Figure 2a - d, Figure S1c, Table S3).” In addition, we included Table S3 to summarize the DAN-c1 labeling patterns observed across these samples.

Weakness #2: A first major weakness is that the main conclusion of the paper, which pertains to associative memory (last sentence of the abstract, and throughout the manuscript), is not justified by their evidence. Why so? Consider the paradigm in Figure 2g, and the data in Figure 2h (22 degrees, the control condition), where the assay and the experimental rationale used throughout the manuscript are introduced. Different groups of larvae are exposed, for 30min, to an odour paired with either (i) quinine solution (red bar), (ii) distilled water (yellow bar, or iii) sucrose solution (blue bar); in all cases this is followed by a choice test for the odour on one side and a distilled-water blank on the other side of a testing Petri dish. The authors observe that odour preference is low after odour-quinine pairing, intermediate after odour-water pairing and high after odour-sucrose pairing. The differences in odour preference relative to the odour-water case are interpreted as reflecting odour-quinine aversive associations and odour-sucrose appetitive associations, respectively. However, these differences could just as well reflect non-associative effects of the 30-min quinine or sucrose exposure per se (for a classical discussion of such types of issues see Rescorla 1988, Annu Rev Neurosci, or regarding Drosophila Tully 1988, Behav Genetics, or with some reference to the original paper by Honjo & Furukubo-Tokunaga 2005, J Neurosci that the authors reference, also Gerber & Stocker 2007, Chem Sens).As it stands, therefore, the current 3-group type of comparison does not allow conclusions about associative learning.

We adopted the single-odor larval learning paradigm from Honjo et al., who first developed and validated this method for studying larval olfactory associative learning7,8. To address the reviewer’s concern regarding potential non-associative effects from 30-minute exposure to quinine or sucrose, we refer to multiple lines of evidence provided in Honjo’s studies: (1) Honjo et al. demonstrated that only larvae receiving paired presentations of odor and unconditioned stimulus (quinine or sucrose) exhibited learned responses. Exposure to either stimulus alone, or temporally dissociated presentations, failed to induce any learning response. (2) When tested with a second, non-trained odorant, larvae only responded to the odorant previously paired with the unconditioned stimulus. This rules out generalized olfactory suppression and confirms odor-specific associative learning. (3) Well-characterized learning mutants (e.g., *rutabaga*, *dunce*) that show deficits in adult reciprocal odor learning also failed to exhibit learned responses in this single-odor paradigm, further supporting its validity. (4) In our study, we used two distinct odorants (pentyl acetate and propionic acid) and two independent D2R knockdown approaches (UAS-miR and UAS-RNAi). We consistently observed that D2R knockdown in DAN-c1 impaired aversive learning. Importantly, naïve olfactory, gustatory, and locomotor assays ruled out general sensory or motor defects. Comparisons with control groups (odor paired with distilled water) also ruled out non-associative effects such as habituation. Taken together, these results strongly support that the single-odor paradigm is a robust and reliable assay for assessing larval olfactory associative learning in *Drosophila*. We have added a section in the Discussion to clarify and defend the use of this paradigm in our study.

Weakness #3: A second major weakness is apparent when considering the sketch in Figure 2g and the equation defining the response index (R.I.) (line 480). The point is that the larvae that are located in the middle zone are not included in the denominator. This can inflate scores and is not appropriate. That is, suppose from a group of 30 animals (line 471) only 1 chooses the odor side and 29, bedazzled after 30-min quinine or sucrose exposure or otherwise confused by a given opto- or thermogenetic treatment, stay in the middle zone... a P.I. of 1.0 would result.

We gave 5 min during the testing stage to allow the larvae to wander on the testing plate. Under most conditions, more than half of larvae (>50%) will explore around, and the rest may stay in the middle zone (will not be calculated). We used 25-50 larvae in each learning assay, so finally around 10-30 larvae will locate in two semicircular areas. Indeed, based on our raw data, a R.I. of 1 seldom appears. Most of the R.I.s fall into a region from -0.2 to 0.8. We should admit that the calculation equation of R. I. is not linear, so it would be sharper (change steeply) when it approaches -1 and 1. However, as most of the values fall into the region from -0.2 to 0.8, we think ‘border effects’ can be neglected if we have enough numbers of larvae in the calculation (10-30).

Weakness #4: Unless experimentally demonstrated, claims that the thermogenetic effector shibire/ts reduces dopamine release from DANs are questionable. This is because firstly, there might be shibire/ts-insensitive ways of dopamine release, and secondly because shibire/ts may affect co-transmitter release from DANs.

*Shibire*^ts1^ gene encodes a thermosensitive mutant of dynamin, expressing this mutant version in target neurons will block neurotransmitter release at the ambient temperature higher than 30C, as it represses vesicle recycling[7]. It is a widely used tool to examine whether the target neuron is involved in a specific physiological function. We cannot rule out that there might be *Shibire*^ts1^ insensitive ways of dopamine release exist. However, blocking dopamine release from DAN-c1 with *Shibire*^ts1^ has already led to learning responses changing (Figure 2h). This result indicated that the dopamine release from DAN-c1 during training is important for larval aversive learning, which has already supported our hypothesis.

For the second question about the potential co-transmitter release, we think it is a great question. Recently Yamazaki et al. reported co-neurotransmitters in dopaminergic system modulate adult olfactory memories in *Drosophila*[9], and we cannot rule out the roles of co-released neurotransmitters/neuropeptides in larval learning. Ideally, if we could observe the real time changes of dopamine release from DAN-c1 in wild type and TH knockdown larvae would answer this question. However, live imaging of dopamine release from one dopaminergic neuron is not practical for us at this time. On the other hand, the roles of dopamine receptors in olfactory associative learning support that dopamine is important for *Drosophila* learning. D1 receptor, dDA1, has been proven to be involved in both adult and larval appetitive and aversive learning[10,11]. In our work, D2R in the mushroom body showed important roles in both larval appetitive and aversive learning (Figure 6a). All this evidence reveals the importance of dopamine in *Drosophila* olfactory associative learning. In addition, there is too much unknow information about the co-release neurotransmitter/neuropeptides, as well as their potential complex ‘interaction/crosstalk’ relations. We believe that investigation of co-released neurotransmitter/neuropeptides is beyond the scope of this study at this time.

Weakness #5: It is not clear whether the genetic controls when using the Gal4/ UAS system are the homozygous, parental strains (XY-Gal4/ XY-Gal4 and UAS-effector/ UAS-effector), or as is standard in the field the heterozygous driver (XY-Gal4/ wildtype) and effector controls (UAS-effector/ wildtype) (in some cases effector controls appear to be missing, e.g. Figure 4d, Figure S4e, Figure S5c).

Almost all controls we used were homozygous parental strains. They did not show abnormal behaviors in either learnings or naïve sensory or locomotion assays. The only exception is the control for DAN-c1, the larvae from homozygous R76F02AD; R55C10DBD strain showed much reduced locomotion speed (Figure S6). To prevent this reduced locomotion speed affecting the learning ability, we used heterozygous R76F02AD; R55C10DBD/wildtype as control, which showed normal learning, naïve sensory and locomotion abilities (Figure 4e to i).

For Figure 4d, it is a column graph to quantify the efficiency of D2R knockdown with miR. Because we need to induce and quantify the knockdown effect in specific DANs (DM1), only TH-GAL4 can be used as the control group, rather than UAS-D2R-miR. For the missing control groups in Figure S4e and S5c, we have shown them in other Figures (Figure 4e).

We described this in the Materials and Methods part, “All control strains used in learning assays were homozygous (except DAN-c1×WT), while all experimental groups (D2R knockdown and thermogenetics) used were heterozygous by crossing the corresponding control strains”.

We also re-organized the Figure S4e and S5c along with the control groups to make it easier to understand.

Weakness #6: As recently suggested by Yamada et al 2024, bioRxiv, high cAMP can lead to synaptic depression (sic). That would call into question the interpretation of low-Dop2R leading to high-cAMP, leading to high-dopamine release, and thus the authors interpretation of the matching effects of low-Dop2R and driving DANs.

We appreciate the reviewer’s suggestion. We read through this literature, which also addresses the question we mentioned in the Discussion section, about the discrepancy between the cAMP elevation in the mushroom body neurons and the reduced MBN-MBON synaptic plasticity after olfactory associative learning in *Drosophila*. The author gave an explanation to the existing D1R-cAMP elevation-MBN-MBON LTD axis, which is really helpful to our understanding about the learning mechanism. However, unfortunately, we do not think this offers a possible explanation for our D2R-related mechanisms. We added this literature into our citation.

**Recommendations for the authors:**

**Reviewer #1 (Recommendations for the authors):**
(1) Throughout the behavioral experiments, a defect in aversive learning is defined as a relative increase in the response index (RI) after olfactory training with quinine (red) and a defect in appetitive learning as a relative decrease in RI after training with sucrose (blue). Training with distilled water (yellow) is intended to be a control for comparisons within genotypes/treatment groups but causes interpretation issues if it is also affected by experimental manipulations.The authors typically make comparisons between quinine, water, and sucrose within each group, but this often forces readers to infer the key comparisons of interest. For example, the key comparison in Figure 2h is the statistically significant difference between the red groups, which differ only in the temperature used during training. Many other figure panels in the paper would also benefit from more direct statistical comparisons, particularly Figure 2k.While I recognize the value of the water control, I strongly recommend that the authors make statistical comparisons directly between genotypes/treatment groups where possible and to interpret results with more caution when the water RI score differs substantially between groups. Also, since the authors are conducting two-way ANOVAs before Dunnett's multiple comparisons tests, they ideally should report the p-value for the main effect of each factor, plus the interaction p-value between the two factors before making multiple comparisons.

We appreciate the reviewer’s suggestion. In response, we re-analyzed all learning assay data in Figures 2 and 4 using two-way ANOVA followed by Tukey’s multiple comparisons test. Unlike our previous analysis, which only compared each experimental group to its corresponding DW control, we now compared all groups against one another. First, we found that most R.I. values from different temperature conditions (Figure 2) or genotypes (Figure 4) trained with DW were not significantly different, with the exception of the data in Figure 2i (formerly Figure 2k; discussed further below). The R.I. from DAN-c1 × D2R-miR larvae trained with QUI was significantly different from both genotype control groups (DAN-c1 × WT and UAS-D2R-miR), while no significant difference was observed between the two controls trained with QUI. Thus, this more comprehensive statistical approach supports the conclusions we previously reported. Second, as the reviewer noted, the new analysis allows for a more direct interpretation of our findings. For example, in the thermogenetic experiments using the Shibire^ts1^ strain, the R.I. of DAN-c1 × UAS-Shibire^ts1^ larvae trained with QUI at 34°C was not significantly different from the DW group at 34°C, but was significantly different from the QUI group at 22°C. Both findings support our conclusion that blocking dopamine release from DAN-c1 impairs larval aversive learning (Figure 2f).

In the dTRPA1 activation experiments, the R.I. of DAN-c1 × UAS-dTRPA1 larvae trained with DW at 34°C was significantly lower than that of the DW group at 22°C and the QUI group at 34°C, but not significantly different from the QUI group at 22°C (Figure 2i). These results indicate that activating DAN-c1 during training is sufficient to drive aversive learning even in the absence of QUI. Interestingly, when DAN-c1 × UAS-dTRPA1 larvae were trained with QUI at 34°C, their R.I. was significantly higher than that of the DW group at 34°C and significantly different from the QUI group at 22°C, but not significantly different from the DW group at 22°C (Figure 2i). We interpret this as evidence that simultaneous activation of DAN-c1 by both QUI and dTRPA1 leads to over-excitation, which in turn impairs aversive learning.

We have revised the figures (Figures 2, 4, 5, and 6) and updated the corresponding Results sections to reflect this new statistical analysis. Additionally, we now report the p-values for interaction, row factor, and column factor - either in Table S4 (for Figure 2) or in the figure captions for Figures 4, 5, 6, S4, S5, and S7.

(2) The authors' motivation to find tools that label DANs other than DAN-c1 was unclear until much later in the paper when I saw the screening experiments in Figures S4 and S5. The authors could provide a clearer justification for why they focus on DAN-c1 in Figure 2 rather than another DAN for which they found a specific driver in Figure 1. The motivation for looking at individual pPAM neurons was also unclear.

We sincerely appreciate the reviewer’s thoughtful suggestion. Our study was initially motivated by the goal of characterizing the expression pattern of D2R in the larval brain. From there, we aimed to identify DAN drivers that label specific pairs of dopaminergic neurons, enabling us to assess the functional role of D2R in distinct DAN subtypes through targeted knockdown experiments. This approach ultimately led us to focus on DAN-c1, as it was the only neuronal population for which D2R knockdown resulted in a learning deficit. We then returned to examine the functional significance of DAN-c1 in aversive learning. While we recognize that a more comprehensive narrative might be desirable, the current structure of our manuscript reflects the most logical progression of our work based on our research priorities and experimental outcomes. We did explore alternative manuscript structures - such as beginning with the D2R expression pattern - but found that the current format best conveys our findings and rtionale.

Regarding our motivation to study individual PAM neurons: we aimed to identify whether D2R plays a role in a specific pair of pPAM neurons involved in larval appetitive learning. However, we were unable to find a driver that exclusively labels DAN-j1, which we believe to be the key neuron in this context (see Figure 1). As a result, our investigation into appetitive learning did not progress beyond the observation of D2R expression in pPAM neurons (Figure 3d), and we did not proceed with learning assays in this context. While we acknowledge the limitations of our study, we believe that our focus on DAN-c1 is well-justified based on both our findings and the tools currently available. We respectfully note that a major restructuring of the manuscript would not necessarily clarify the rationale for focusing on DAN-c1, and therefore we have maintained the current organization.

(3) The authors should also double-check and update the expression patterns of the drivers in Table 1 using references such as the FlyLight online resource. For example, MB438B labels PPL1-α'2α2, PPL1-α3, PPL1-γ1pedc according to FlyLight, not just PPL1-γ1pedc as initially reported by Aso and Hattori et al. (2014).

We appreciate the reviewer’s suggestion. We have double-checked and updated the driver expression patterns in Table 1, using FlyLight data as a reference.

(4) Interpreting overlaid green-and-red fluorescence confocal images would be difficult for any colorblind readers; I suggest that the authors consider using a more friendly color set.

We thank the reviewer for the suggestion. In our study, we need three distinct colors to represent different channels. We also tested an alternative color scheme using and cyan , magenta, and yellow (CMY) instead of the standard red, green, and blue (RGB). As a comparison (see below), we used a R76F02AD;R55C10DBD (DAN-c1) GFP-labeled brain as an example. In our evaluation, the RGB combination provided clearer visualization and appeared more natural, while the CMY scheme looked somewhat artificial. Therefore, we decided to retain the original RGB color scheme and did not modify the colors in the figures.

(5) For Figure 4d, counting each DAN as an individual N would violate the assumption of independence made by the unpaired t test, since multiple DANs are found in each brain and therefore are not independent. Instead, it would be better to count each individual N as the average intensity of the four DANs measured in each brain.

We revised the analysis of microRNA efficiency by averaging the fluorescence intensity of DANs within each brain, treating each brain as a single sample. Based on this approach, we re-plotted Figure 4d.

(6) Finally, the authors ought to make it clearer throughout the paper that they have implicated a pair of DAN-c1 neurons in aversive learning, not just a single DAN as currently stated in the title.

We thank the reviewer for the suggestion about the phrase we are using under this scenario. We have changed all “single neuron” to “a pair of neurons”.

**Reviewer #2 (Recommendations for the authors):**
(1) The results section presents: "Activation of DAN-c1 with dTRPA1 at 34°C during training induced repulsion to PA in the distilled water group (Figure 2k). These data suggested that DAN-c1 excitation and presumably increased dopamine release is sufficient for larval aversive learning in the absence of gustatory pairing."An alternative interpretation is that 30 min of TrpA activation depletes synaptic vesicle pool, or inactivates neurons because of prolonged depolarization, or DAN shows firing rate adaptation (e.g. see Pulver et al. 2009; doi:10.1152/jn.00071.2009). In such a case DA release would be reduced and not increased. Therefore, the interpretation that DAN-c1 activation is both necessary and sufficient in larval aversive learning is difficult to be sustained.In this regard it is important to know how the sensory motor abilities are during a thermos-induction at 34°C during 30 min.

We thank the reviewer for the thoughtful suggestion. Regarding the concern about potential dopamine depletion or neuronal inactivation, we believe a comparison with the Shibire^ts1^ experiments helps clarify the interpretation. Activation of Shibire^ts1^ during training with distilled water did not result in aversive learning (Figure 2f), which is a distinct phenotype from that observed with dTRPA1 activation (Figure 2i). This suggests that the phenotypes seen with dTRPA1 activation are not due to reduced dopamine release. Additionally, as the reviewer suggested, we have revised our conclusion to state that “DAN-c1 is important for larval aversive learning,” rather than claiming it is both necessary and sufficient.

(2) The GRASP system can label the contact of a cell in close proximity like synaptic contacts, but also other situations like no synaptic contact. It would be useful to use a more specific synaptic labelling tool, like the trans-synaptic tracing system (Talay et al., 2017 https://doi.org/10.1016/j.neuron.2017.10.011), which provides a better label of synaptic contact.

We really appreciate the reviewer’s suggestion. First, we acknowledge that there are four general methods to reveal synaptic connections between neurons: immunohistochemistry (IHC), neuron labeling, viral tracing, GRASP, and electron microscopy (EM). Among these, IHC is not sufficiently convincing, viral tracing is challenging and rarely used in Drosophila, and EM, while the most accurate, is prohibitively expensive for our current goals. For these reasons, we chose the GRASP system to demonstrate the synaptic connections from dopaminergic neurons to the mushroom body. Second, we utilized an activity-dependent version of the GRASP system, linking split-GFP1-10 with synaptic proteins (e.g., synaptobrevin)[12] rather than with cell surface proteins like CD4 or CD8. This version significantly reduces false positive signals compared to the previous version, which was tagged with cell surface proteins. While we admit that this method does not provide as solid evidence of synaptic connections as EM, it is the most efficient method available to us for showing the synaptic connections from dopaminergic neurons to the mushroom body. Finally, we thank the reviewer for suggesting the literature on trans-synaptic tracing methods. Unfortunately, this method is not suitable for our goal, as it labels the entire postsynaptic neuron. In our study, we use GRASP to identify the specific dopaminergic neurons based on the synaptic locations and compartments within the mushroom body lobe. We require a labeling system at the subcellular level because, as noted, DAN-c1 forms synapses specifically in the lower peduncle (LP) of the mushroom body lobe, which is part of the axonal bundles from mushroom body neurons. Using the trans-synaptic tracing method would label the entire mushroom body, making it impossible to distinguish DAN-c1 from other DL1 dopaminergic neurons.

(3) Previously, Honjo et al (2009) used a petri dish of 8.5 cm and a filter paper for reinforcement of 5.5 cm. In this study the petri dish was 10 cm and the size of the filter paper was not informed. That is important information because it will determine the probability of conditioning.

A piece of filter paper (0.25cm^2^ square) was used to hold odorants in this study. We have added this information to the Materials and Methods.

(4) Statistic analysis of Behavioral performance of Fig 2H-I was made by ANOVA followed by Dunnett multiple comparisons test. Which was the control group? In each graph 2 independent Dunnett tests were performed against the DW control group?

We have re-analyzed the data using a two-way ANOVA followed by Tukey’s multiple comparison test, as suggested by Reviewer #1. In Figure 2f-j (previously Figure 2h-l), the DW groups serve as the control groups. In our new analysis, we compared data across all groups using Tukey’s multiple comparison test, with particular focus on comparisons to the corresponding DW control groups.

(5) The sample size in staining experiments of figures 1-4 were not informed.

We have added Table S2 in the supplementary materials to provide the N numbers for brain samples used in the figures.

(6) Color code in Fig 5 is missing, I assumed that is the same as in figure 4e

We added color code in the figure legend of Figure 5.

(7) Line 506 "0.1% QH solutions" should be 0.1% QUI solutions

Changed.

(8) There is no information on the availability of data

We added Data Availability Statement: Data will be made available on request.

**Reviewer #3 (Recommendations for the authors):**
(1) Axes of behavioural experiments should better show the full span of possible values (-1;1) to allow a fair assessment.

We have adjusted the axes in all learning assay graphs to a range from -1 to 1 for consistency and clarity.

(2) Ns should better be given within the figures.

We have added Table S2 in the supplementary materials to provide the N numbers for brain samples used in the figures. Additionally, Tables S4 to S6 include the N numbers for the learning assays. While we initially considered including the N numbers within the figure captions, we found it challenging to present this information clearly and efficiently. Therefore, we decided to summarize the N numbers in the tables instead.

(3) Dot- or box-plots would be better for visualizing the data than means and SEMs.

We agree with the reviewer’s suggestion. In the behavioral assay graphs, both dot plots and mean ± SEM have been included for better visualization of the data.

(4) The paper reads as if Dop2R would reduce neuronal activity, rather than "just" cAMP levels. Such a misunderstanding should be avoided.

We appreciate the reviewer’s comment. Under most conditions, dopamine binding to D2Rs activates the Gαi/o pathway, which inhibits adenylyl cyclase (AC) and reduces cAMP levels. This reduction in cAMP ultimately leads to decreased neuronal activity. In other words, D2R activation typically has an inhibitory effect on neurons. Additionally, D2R can exert inhibitory effects through other signaling pathways, such as the inhibition of voltage-gated associative learning, we continue to emphasize the importance of the D2R-mediated AC-cAMP-PKA signaling pathway. However, we do not rule out the potential involvement of additional signaling pathways, such as inhibition of voltage-gated calcium channels via Gβγ subunits[5]. As noted in the Introduction, dopamine receptors are also involved in other signaling cascades, including PKC, MAPK, and CaMKII pathways. In the context of our study, based on current understanding of molecular signaling in Drosophila olfactory, we still think D2R mediated AC-cAMP-PKA signaling pathway would be the most important one. However, we cannot rule out the involvement of other signaling pathways.

(5) It would be better if citations were more clearly separated into ones that refer to adult flies versus work on larvae.

We separated the citations related to adult flies from those working on larvae.

(6) Line 81-83. DopECR is not found in mammals, is it?

You are correct. DopECR is not found in mammals. This non-canonical receptor shares structural homology with vertebrate β-adrenergic-like receptors. It can be activated rapidly by dopamine as well as insect ecdysteroids[13,14].

(7) Line 99: Better "a" learning center (some forms of learning work without mushroom bodies).

We have revised the text from "the learning center" to "a learning center," as suggested by the reviewer.

(8) Supplemental figures should be numbered according to the sequence in which they are mentioned in the text.

We have rearranged the sequence of supplemental figures to match the order in which they are referenced in the text.

(9) It is striking that dTRPA1-driving DANc1 is punishing in the water condition but that this effect does not summate with quinine punishment (but rather seems to impair it). Maybe you can back this up by ChR- or Chrimson-driving DANc1? Or by silencing DANc1 by GtACR1?

We appreciate the reviewer’s suggestion. Indeed, we observed similar but not identical results when we used ChR2 to activate DAN-c1 during the training stage (Figure 5b and c). We found that activating DAN-c1 with quinine (QUI) impaired aversive learning (Figure 5b), consistent with our findings using dTRPA1 activation of DAN-c1 when trained in QUI at 34°C (Figure 2i). We propose that the over-excitation of DAN-c1, whether induced by QUI or artificial manipulation (optogenetics and thermogenetics), impairs aversive learning, which aligns with our findings for D2R knockdown (Figure 4e). However, there are some differences between dTRPA1 and ChR2 activation. While dTRPA1 activation induced aversive learning when trained with distilled water (DW) at 34°C (Figure 2i), ChR2 did not induce aversive learning under the same conditions (Figure 5c). We believe this difference is due to the varying activation levels between the two manipulations. Our optogenetic stimulus may have been stronger than the thermogenetic one, potentially leading to over-excitation in the DW group, preventing aversive learning. In the QUI group, the more severe over-excitation impaired aversive learning, producing a phenotype similar to that observed with other over-excitation methods (e.g., thermogenetics or D2R knockdown), where the phenotype reached a maximum level. We have also addressed these points in the Discussion section.

(10) Unless I got the experimental procedure wrong, isn't it surprising that Figure S7b does not uncover a punishing effect of driving TH-Gals neurons?

This optogenetic experiment with ChR2 expression in TH-GAL4 neurons was a pioneering attempt to activate DAN-c1 using ChR2. As explained in response to question (9), the failure to observe a punishing effect in the DW group when TH-GAL4 neurons were activated during training may be due to our optogenetic stimulus being too strong. This likely resulted in over-excitation of DAN-c1 (among the neurons labeled by TH-GAL4), impairing aversive learning and preventing the appearance of typical aversive behaviors.

(11) It seems that Figure1f´ is repeated, in a mirrored manner, in Figure 2e.

We have removed Figure 2e, as it was deemed redundant and not necessary for this section.

Reference

(1) Saumweber, T. et al. Functional architecture of reward learning in mushroom body extrinsic neurons of larval Drosophila. Nat Commun 9, 1104 (2018). https://doi.org/10.1038/s41467-018-03130-1

(2) Aso, Y. & Rubin, G. M. Dopaminergic neurons write and update memories with cell-type-specific rules. Elife 5 (2016). https://doi.org/10.7554/eLife.16135

(3) Xie, T. et al. A Genetic Toolkit for Dissecting Dopamine Circuit Function in Drosophila. Cell Rep 23, 652-665 (2018). https://doi.org/10.1016/j.celrep.2018.03.068

(4) Eschbach, C. et al. Recurrent architecture for adaptive regulation of learning in the insect brain. Nat Neurosci 23, 544-555 (2020). https://doi.org/10.1038/s41593-020-0607-9

(5) Neve, K. A., Seamans, J. K. & Trantham-Davidson, H. Dopamine receptor signaling. J Recept Signal Transduct Res 24, 165-205 (2004). https://doi.org/10.1081/rrs-200029981

(6) Draper, I., Kurshan, P. T., McBride, E., Jackson, F. R. & Kopin, A. S. Locomotor activity is regulated by D2-like receptors in Drosophila: an anatomic and functional analysis. Dev Neurobiol 67, 378-393 (2007). https://doi.org/10.1002/dneu.20355

(7) Honjo, K. & Furukubo-Tokunaga, K. Induction of cAMP response element-binding protein-dependent medium-term memory by appetitive gustatory reinforcement in Drosophila larvae. J Neurosci 25, 7905-7913 (2005). https://doi.org/10.1523/JNEUROSCI.2135-05.2005

(8) Honjo, K. & Furukubo-Tokunaga, K. Distinctive neuronal networks and biochemical pathways for appetitive and aversive memory in Drosophila larvae. J Neurosci 29, 852-862 (2009). https://doi.org/10.1523/JNEUROSCI.1315-08.2009

(9) Yamazaki, D., Maeyama, Y. & Tabata, T. Combinatory Actions of Co-transmitters in Dopaminergic Systems Modulate Drosophila Olfactory Memories. J Neurosci 43, 8294-8305 (2023). https://doi.org/10.1523/jneurosci.2152-22.2023

(10) Selcho, M., Pauls, D., Han, K. A., Stocker, R. F. & Thum, A. S. The role of dopamine in Drosophila larval classical olfactory conditioning. PLoS One 4, e5897 (2009). https://doi.org/10.1371/journal.pone.0005897

(11) Kim, Y. C., Lee, H. G. & Han, K. A. D1 dopamine receptor dDA1 is required in the mushroom body neurons for aversive and appetitive learning in Drosophila. J Neurosci 27, 7640-7647 (2007). https://doi.org/10.1523/JNEUROSCI.1167-07.2007

(12) Macpherson, L. J. et al. Dynamic labelling of neural connections in multiple colours by trans-synaptic fluorescence complementation. Nat Commun 6, 10024 (2015). https://doi.org/10.1038/ncomms10024

(13) Abrieux, A., Duportets, L., Debernard, S., Gadenne, C. & Anton, S. The GPCR membrane receptor, DopEcR, mediates the actions of both dopamine and ecdysone to control sex pheromone perception in an insect. Front Behav Neurosci 8, 312 (2014). https://doi.org/10.3389/fnbeh.2014.00312

(14) Lark, A., Kitamoto, T. & Martin, J. R. Modulation of neuronal activity in the Drosophila mushroom body by DopEcR, a unique dual receptor for ecdysone and dopamine. Biochim Biophys Acta Mol Cell Res 1864, 1578-1588 (2017). https://doi.org/10.1016/j.bbamcr.2017.05.015